# Morphological Characteristics and Molecular Evidence Reveal four New Species of *Russula* subg. *Brevipedum* from China

**DOI:** 10.3390/jof9010061

**Published:** 2022-12-30

**Authors:** Yanliu Chen, Mengya An, Jingying Liang, Weijie Li, Chunying Deng, Jing Wang, Yangkun Li, Junfeng Liang

**Affiliations:** 1Research Institute of Tropical Forestry, Chinese Academy of Forestry, Guangzhou 510520, China; 2College of Forestry, Nanjing Forestry University, Nanjing 210037, China; 3Nanjing University of Chinese Medicine, Nanjing 210023, China; 4Guizhou Institute of Biology, Guizhou Academy of Science, Guiyang 550003, China; 5DNADecode (Tianjin) Life Technology Co., Ltd., Tianjin 300350, China

**Keywords:** agaricomycetes, diversity, phylogeny, morphology, taxonomy

## Abstract

In this study, four new species of *Russula* subg. *Brevipedum* collected from China are described based on morphological characteristics and molecular evidence. *Russula brevispora* has a white body whose whole parts often stain brownish-orange or grayish-orange, extremely crowded lamellae with the presence of lamellulae, basidiospores with low warts and an inamyloid suprahilar spot, and clavate or lageniform hymenial cystidia often with a papillary or submoniliform appendage. *Russula flavescens* is characterized by a white pileus often turning yellowish brown when touched, white lamellae turning brown or light orange, basidiospores with an inamyloid suprahilar spot, and fusiform hymenial cystidia often with an appendage. *Russula longicollis* is morphologically characterized by a white pileus, turning grayish orange when bruised, white lamellae turning pale yellow when bruised, basidiospores with isolated warts and an amyloid suprahilar spot, and fusiform hymenial cystidia usually with a long appendage. *Russula pseudojaponica* has a yellowish-brown pileus center, yellowish lamellae unchanging when bruised, basidiospores with low warts and an inamyloid suprahilar spot, clavate hymenial cystidia often with a papillary appendage, and clavate pileocystidia with granulose contents. Phylogenetic analyses showed that *R. flavescens*, *R. brevispora*, and *R. pseudojaponica* are members of the subsect. *Pallidosporinae*, whereas *R. longicollis* belongs to subsect. *Lactarioideae*, and is somewhat related to *R. leucocarpa*.

## 1. Introduction

*Russula* Pers. is a genus with a great diversity of mushroom-forming fungi species, predictably containing at least 2000 species within the genus [1,2]. The members of this genus are important plant root symbionts in forest ecosystems, occurring across a wide range of habitats, from the arctic tundra to tropical forests [3]. Recently, *Russula* has been classified into eight subgenera: *Russula* subg. *Archaea* (Buyck and V. Hofst.), *R.* subg. *Brevipedum* (Buyck and V. Hofst.), *R.* subg. *Compactae* (Fr.) (Bon), *R.* subg. *Crassotunicata* (Buyck and V. Hofst.), *R.* subg. *Glutinosae* (Buyck and X.H. Wang), *R.* subg. *Heterophyllidiae* (Romagn.), *R.* subg. *Malodorae* (Buyck and V. Hofst.), and *R.* subg. *Russula* [4,5].

The *Russula* subg. *Brevipedum*, typified by *R. brevipes* Peck, was originally described in 2015 as *R.* subg. *Brevipes*; however, this was an invalid name and was changed to *Brevipedum* in 2020 [5]. The members of this subgenus mostly have a medium to very large basidiomata, which often stains yellowish-brown to reddish-brown, regularly unequal lamellae, a distinct smell or acrid to strongly acrid taste, a whitish to yellow spore print, and mucronate to obtuse-rounded cystidia in all parts of the fruiting body [4]. The species of the *R.* subg. *Brevipedum* was initially divided into two subsections: *R.* subsect. *Lactarioideae* Maire (*R.* subsect. *Delicinae* Bat), and *R.* subsect. *Pallidosporinae* Bon, a division subsequently supported by DNA analyses [6]. The *Russula* subsect. *Lactarioideae* (type *R. delica* Fr.) was once mistaken for the genus *Lactifluus* (Pers.) Roussel because of its white milky pileus [7]. In fact, it is distinguished without difficulty by the combination of thick fleshy basidiocarps, a whitish pileus and stipe staining yellowish-brown when mature, and the regularly unequal, whitish to greenish-white or pale yellowish lamellae [8,9]. *Russula* subsect. *Pallidosporinae* was originally mentioned by Bon [10]. It is characterized by an ochraceous cream to yellow spore print, yellowish lamellae, basidiospores with inamyloid suprahilar, and a distinctly thicker pileipellis. This subgenus has a cosmopolitan distribution, in Europe [10,11,12], North and South America [8,13,14,15,16], Australia [17,18], Africa [19], and Asia [20,21,22,23]. In China, a total of 13 taxa of *R.* subg. *Brevipedum* have been reported [9,24,25,26,27].

*Russula japonica* Hongo is a poisonous species in *R.* subg. *Brevipedum* originally described from Japan with a white body, crowded lamellae, and a stumpy stipe [28]. In the past, some Chinese *Russula* specimens with similar characteristics were reported as *R. japonica*, especially specimens with gastrointestinal toxicity [29,30,31,32,33,34,35]. However, morphological characteristics and molecular evidence showed that these specimens were different from *R*. *japonica*. Based on comparative studies of available collections, four new species of *R.* subg. *Brevipedum* are proposed, with detailed morphological descriptions coupled with illustrations and phylogenetic analysis.

## 2. Materials and Methods

### 2.1. Morphological Study

Fresh basidiomata from five provinces of China were collected and photographed. The samples were dried at 50 °C and deposited in the herbarium of the Research Institute of Tropical Forestry, Chinese Academy of Forestry (RITF). Macromorphological descriptions were based on detailed notes and photographs. Color codes and terms follow the Methuen Handbook of Colour [36]. Micromorphological characteristics were observed using a ZEISS Imager M2 (Carl Zeiss AG; Oberkohen, Germany) with oil-immersion lenses at a magnification of 1000×. Macroscopic characters were observed in 5% KOH, 1% phloxin, and 1% ammoniacal Congo red. Tissues were mounted in cresyl blue [37], sulfovanillin [1], and treated with carbolfuchsin [38] to observe the presence and color changes of the incrustations and cystidium contents. Basidiospores were observed in Melzer’s reagent and measured in lateral view excluding the height of ornamentations. Basidiospores measurements are represented as (Min–)AV-SD–AV–AV+SD(–Max), where Min is the minimum value, Max is the maximum value, AV is the average value, SD is the standard deviation, and Q represents the length/width ratio of the basidiospores [2]. Basidium length excludes the length of the sterigmata. Hyphal terminations and pileocystidia were observed both near the pileus margin and in the pileus center. Scanning electron microscopy (SEMJEOL JSM-6510) was used to photograph the shape and ornamentation of the basidiospores.

### 2.2. DNA Extraction, PCR, and Sequencing

Total genomic DNA was extracted from the dry specimens following an improved CTAB protocol [39]. The ITS region of rDNA was amplified with the primers ITS1 and ITS4 [40]. The amplification protocol consists of a 5 min initial denaturation at 95 °C, followed by 35 cycles of 30 s at 95 °C, 30 s at 53 °C, and 2 min at 72 °C, with a final extension of 10 min at 72 °C. The remaining three loci (nrLSU, *RPB2*, and mtSSU) were amplified using the primers and protocols complying with Buyck [4]. The PCR products were purified using a TaKaRa MiniBEST Agarose Gel DNA Extraction Kit according to the operation manual. The amplified PCR products were subsequently sequenced on an ABI 3730 DNA analyzer using an ABI BigDye Terminator v3.1 Cycle Sequencing Kit (Shanghai Songon Biological Engineering Technology and Services Co., Ltd., Shanghai, China). The newly generated sequences were submitted to the GenBank database (https://www.ncbi.nlm.nih.gov/genbank, URL (29/9/23) Table 1).

### 2.3. Sequence Alignment and Molecular Phylogenetic Analyses

All the available sequences of species in the subg. *Brevipedum* from GenBank were included for phylogenetic analyses. The original bidirectional sequences were assembled with the help of Contigexpress software [41]. The sequences were aligned using the MAFFT 7.0 online version [42] and manually adjusted in BioEdit 7.0.9 [43]. The four datasets were concatenated using Phyutility v2.2 for further analysis [44]. The final aligned result was submitted to TreeBase (S29992). Maximum likelihood (ML) analysis was carried out in RAxML 7.0.3 [45]. All parameters were kept at the default settings, except the models set as GTRGAMMA, and statistical support was obtained using nonparametric bootstrapping with 1000 replicates. Bootstrap support (BS) greater than 75% was considered significant. Bayesian inference (BI) analysis was conducted in MrBayes 3.2.6 [46]. The best-fit model was estimated with MrModelTest v. 2.3 [47] using the Akaike information criterion (AIC). Four chains were run for 5 million generations sampling from the posterior distribution every 100 generations. Other parameters were kept at the default settings. The analysis was terminated when the average standard deviation of split frequencies was stable below 0.01. The parameters and tree samples were then summarized, and the Bayesian posterior probabilities (BPPs) were calculated after discarding the first 25% of the samples as the burn-in. BPPs over 0.95 were considered significant.

The combined dataset included sequences from 73 specimens representing 35 taxa. The dataset had an aligned length of 3018 characters including gaps, of which ITS contains 772 characters, nrLSU contains 896 characters, mtSSU contains 564 characters, and *RPB2* contains 786 characters. The best model for the Bayesian analysis of the nrLSU and mtSSU sequence datasets was GTR+I+G; for ITS, the best model was HKY+I+G, and for *RPB2* it was SYM+I. Six partitions were implemented for further phylogenetic analyses (ITS, nrLSU, mtSSU, *RPB2* 1st, *RPB2* 2nd, and *RPB2* 3rd).

## 3. Results

### 3.1. Phylogeny

The phylogenetic trees generated from the RAxML and Bayesian analyses were similar in topology; thus, only the ML tree is shown in Figure 1. The phylogenetic analyses confirmed that *R.* subg. *Brevipedum* formed an independent clade with strong support (BS/BPP = 100/1), and can be divided into two lineages, subsect. *Pallidosporinae* and subsect. *Lactarioideae*.

The sequences of the four new species, *R. brevispora*, *R. flavescens*, *R. longicollis*, and *R. pseudojaponica* each formed a strongly supported clade, and were clearly distinct from other known and sequenced species of the subg. *Brevipedum*. *Russula brevispora*, *R. flavescens*, and *R. pseudojaponica* clustered together with *R. vesicatoria* Murrill and *R. japonica* Hongo (BS/BPP = 100%/1. Figure 1). Another species, *R. longicollis***,** was close to *R. leucocarpa* (G.J. Li and C.Y. Deng).

### 3.2. Taxonomy

#### 3.2.1. *Russula brevispora* (Y.L. Chen and J.F. Liang sp. nov.)

Mycobank. MB 846486

Figure 2A,B, Figure 3A,B, Figure 4 and Figure 5

Diagnosis. Differs from all known members (except *R. japonica*) of the subsect. *Pallidosporinae* in shorter basidiospores. It differs from *R. japonica* in its lageniform hymenial cystidia and thinner basidia.

Holotype. CHINA, Guizhou Province, Tongren City, Fanjingshan National Nature Reserve, 27°54′43.25″ N, 108°39′25.62″ E, 1950 m asl., 1 August 2019, C.Y. Deng (HGAS-MF009915), GenBank: MN648956 (ITS); OP850852 (nrLSU); OP856542 (mtSSU).

Etymology. “*brevispora*” refers to shorter basidiospores.

Basidiomata (Figure 2A,B) medium- to large-sized; pileus 60–105 mm in diameter; hemispherical when young, then convex, applanate with a shallowly depressed center when mature; surface smooth, dry, white (1A1), often staining orange white (5A2) or grayish orange (6B5) to brown (6D7); margin not cracked, nonstriate. Lamellae adnate to slightly decurrent, 24–26 pieces at 1 cm near the pileus margin, white (1A1) to cream, often staining grayish-orange (5B4) to brownish-orange (6C8) when bruised; lamellulae frequently present and irregular in length, with furcations absent; edge entire and concolorous. Stipe central, 15–45 × 20–35 mm, cylindrical, slightly tapering towards the base, smooth, white (1A1), solid. Context white (1A1), staining grayish-orange (6B5) when bruised, compact; taste slightly acid, odor inconspicuous. Spore print cream.

**Figure 1 jof-09-00061-f001:**
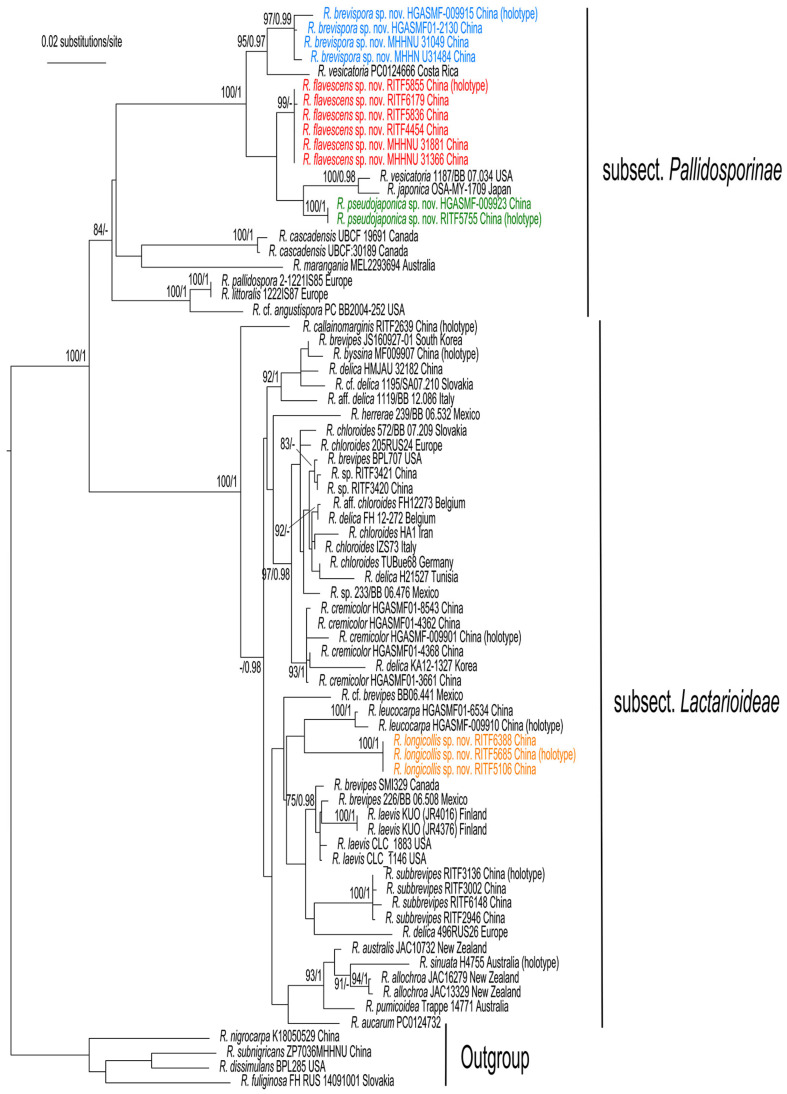
Phylogenetic tree of *R.* subg. *Brevipedum* based on the ITS-nrLSU-*RPB2*-mtSSU dataset. Bootstrap support (BS) ≥ 75% and Bayesian posterior probabilities (PP) ≥ 0.95 are shown. Four species of *R.* subg. *Compactae* were selected as the outgroup. New species are marked with colored characters.

**Figure 2 jof-09-00061-f002:**
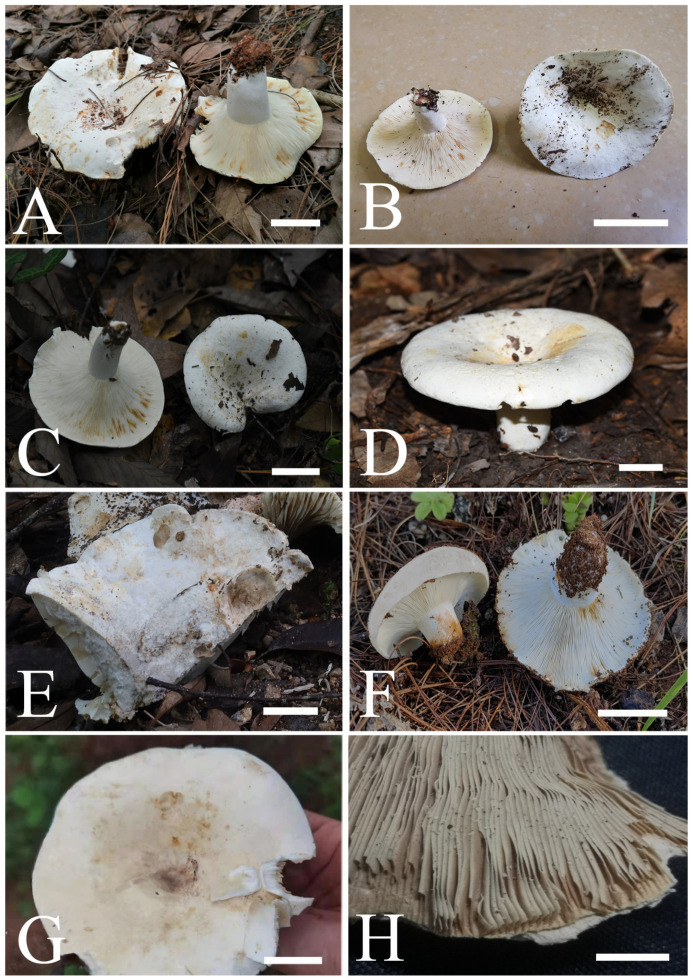
Fruiting bodies of four new species: (**A**) *R. brevispora* (MHHNU 31049, photo by Z.H. Chen); (**B**) *R. brevispora* (MHHNU 31484, photo by Z.H. Chen) (**C**) *R. flavescens* (RITF6179, photo by W.J. Li); (**D**) *R. flavescens* (RITF5855, photo by W.J. Li); (**E**) *R. longicollis* (RITF5685, holotype, photo by Y.L. Chen); (**F**) *R. longicollis* (RITF6388, photo by X.L. Gao); (**G**) *R. pseudojaponica* (RITF5755, holotype, photo by Y.L. Chen); (**H**) *R. pseudojaponica* (HGASMF-009923, dry material, photo by Y.L. Chen). Scale bars = 20 mm.

**Figure 3 jof-09-00061-f003:**
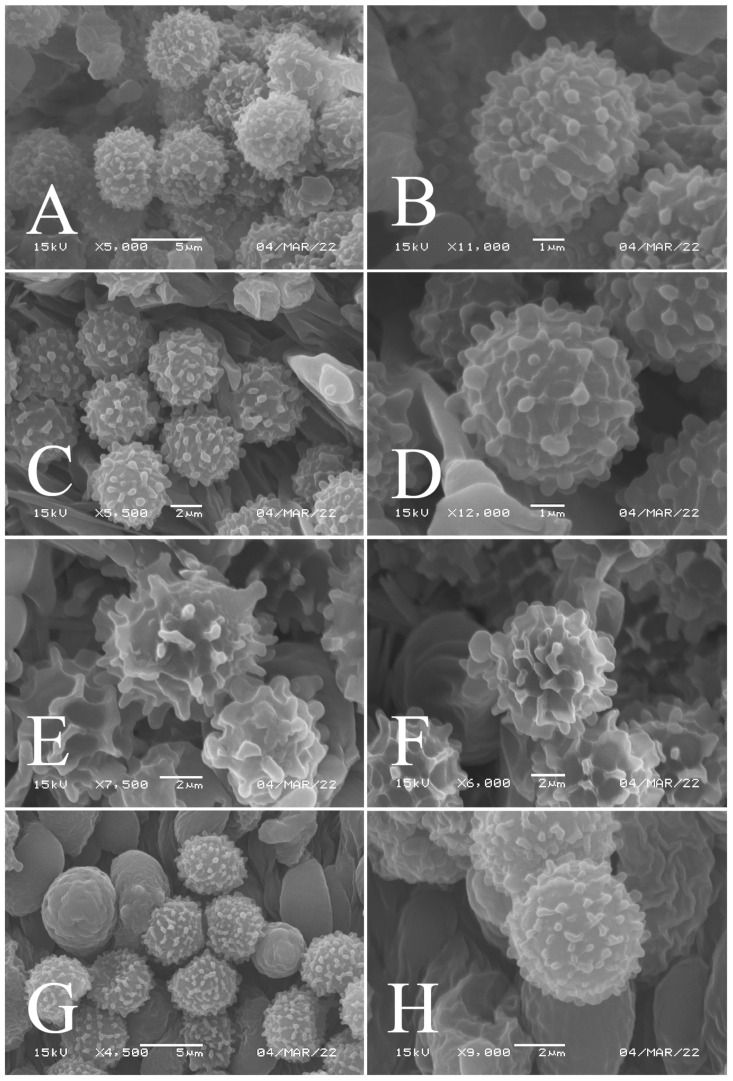
Basidiospores of four new species in SEM: (**A**,**B**) *R. brevispora* (HGASMF-009915, holotype); (**C**,**D**) *R. flavescens* (RITF5855, holotype); (**E**,**F**) *R. longicollis* (RITF5685, holotype); (**G**,**H**) *R. pseudojaponica* (RITF5755, holotype).

**Figure 4 jof-09-00061-f004:**
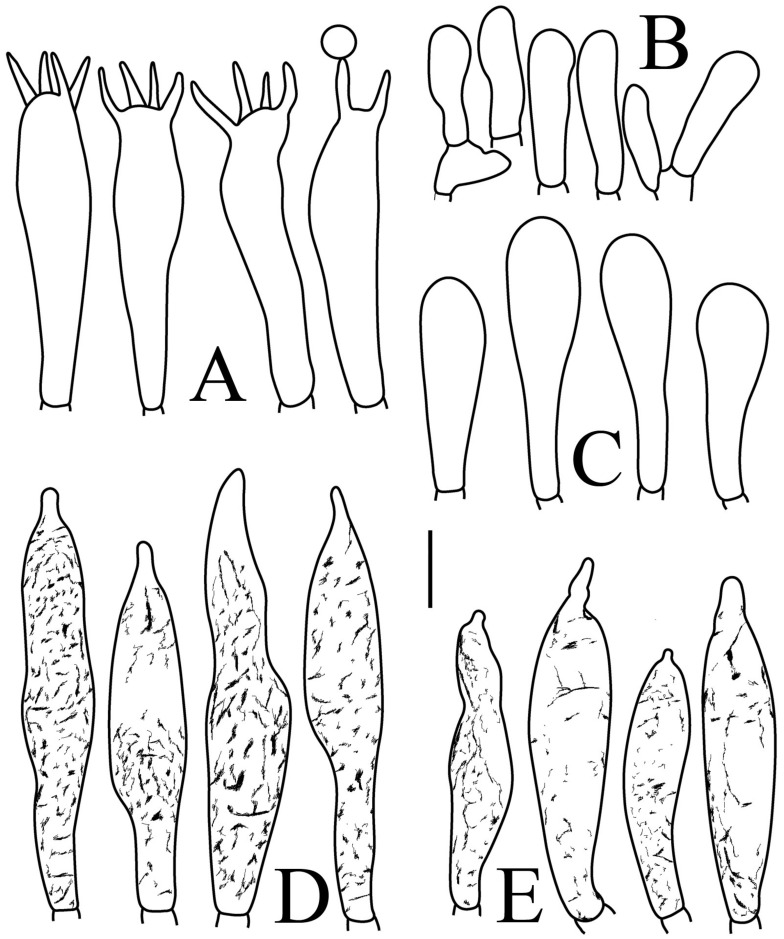
*Russula brevispora.* (HGASMF-009915, holotype): (**A**) basidia; (**B**) marginal cells; (**C**) basidiola; (**D**) hymenial cystidia on lamellae sides; (**E**) hymenial cystidia on lamellae edges. Scale bars: 10 μm.

**Figure 5 jof-09-00061-f005:**
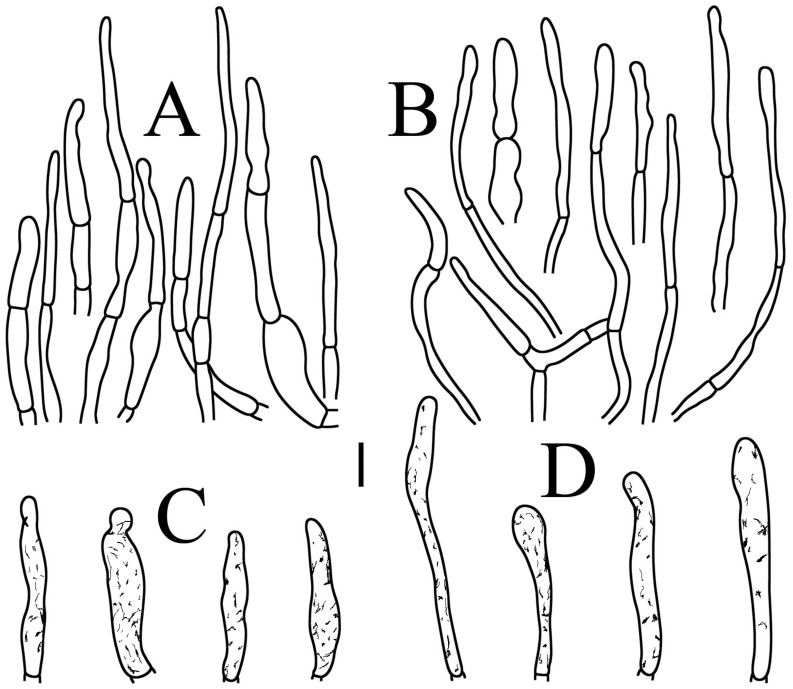
*Russula brevispora.* (HGASMF-009915, holotype): (**A**) hyphal terminations near the pileus margin; (**B**) hyphal terminations near the pileus center; (**C**) pileocystidia near the pileus margin; (**D**) pileocystidia near the pileus center. Scale bars: 10 μm.

Basidiospores (Figure 3A,B) (5.3–)5.6–5.9–6.2(–6.6) × (4.8–)5–5.3–5.6(–6) µm, Q = (1.01–)1.05–1.11–1.17(–1.28), mainly subglobose, sometimes broadly ellipsoid and globose; ornamentation of small to medium-sized, dense to very dense (6–10 in a 3 µm diameter circle) amyloid conical warts, 0.4–0.7 µm high, subreticulate, never fused by fusions, often connected by fine line connections (2–4 in the circle); suprahilar spot small-sized, inamyloid. Basidia (Figure 4A) (31–)34–38–42(–45) × (7.3–)8–8.7–9.3(–9.9) µm, 2–4-spored, narrowly clavate, thin-walled; sterigmata up to 8.1 µm long; basidiola (Figure 4C) clavate, ca. 7.1–9.7 µm wide. Hymenial cystidia on lamellae sides (Figure 4D) moderately numerous, ca. 1200/mm^2^, (39.5–)42.7–51.4–60(–67.5) × (7.4–)8.4–9.3–10.2(–11.3) µm, subfusiform or lageniform, occasionally clavate, apically acute, occasionally obtuse, often with a papillary or coracoid appendage, thin-walled; contents granulose-crystalline or banded, no reaction in SV. Hymenial cystidia on lamellae edges (Figure 4E) dispersed to moderately numerous, ca. 500–900/mm^2^, (37.5–)41–47–53(–59.5) × (7.6–)8.3–9.2–10.1(–11.5) µm, lageniform or clavate, apically obtuse or acute, often with a papillary or submoniliform appendage, thin-walled; contents granulose-crystalline or banded, no reaction in SV. Marginal cells (Figure 4B) (15.3–)16.6–18.2–19.9(–20.2) × (4.8–)5–5.6–6.1(–6.4) µm, subclavate or cylindrical, colorless. Pileipellis orthochromatic in cresyl blue, sharply delimited from the underlying context, two-layered, composed of suprapellis and subpellis; suprapellis 53–100 µm deep, composed of prostrate to ascending hyphae; subpellis composed of closely interweaved colorless hyphae 1.7–4.3 µm wide. Hyphal terminations near the pileus margin (Figure 5A) unbranched, sometimes flexuous, thin-walled, two forms: the first form is usually shorter and broader, terminal cells (17.4–)21.6–26.3–31 (–34.6) × (3–)3.5–4.1–4.7(–5), cylindrical, sometimes apically constricted; the second form is longer and slender, terminal cells (30.2–)34.2–42.7–51.2(–62.8) × (2.4–)3.5–3.1–3.9(–5.1), subconical, cylindrical and apically constricted; the subterminal cells are cylindrical, often expanded or inflated, ca. 3–6.3 µm wide, unbranched. Hyphal terminations near the pileus center (Figure 5B) are sometimes flexuous, occasionally branched, thin-walled, similarly two forms; one-form terminal cells (21.3–)22–24.3–26.5(–28.6) × (3.6–)3.7–4.4–5.2(–5.7) µm, cylindrical, apically obtuse; other form (27.4–)34.9–43.9–52.8(–60) × (2.3–)2.7–3.3–4(–4.7) µm, conical, cylindrical, and apically constricted and attenuated; terminal cells cylindrical or inflated, ca. 2.5–5.6 µm wide, unbranched. Pileocystidia near the pileus margin (Figure 5C) always one-celled, dispersed, (36.3–)38.1–42.3–44.5(–48.1) × 5.1–6.5–8(–9.4) µm, lageniform or fusiform, apically obtuse, sometimes with a papillary appendage, thin-walled; contents granulose-crystalline, no reaction in SV. Pileocystidia near the pileus center (Figure 5D) longer than those near the margin, 40–49.4–57.8(–63.1) × 5.5–6.6–7.7(–8.7) µm, clavate, lageniform or fusiform, apically obtuse, thin-walled; contents granulose-crystalline, no reaction in SV. Cystidioid hyphae dispersed in subpellis and context with granulose contents, oleiferous hyphae frequent in subpellis and context with refringent contents.

Habitat. On the ground in a mixed forest of *Pinus massoniana* Lamb. and *Quercus* sp.

Additional specimens examined. CHINA, Guizhou Province, Qiannan Buyei and Miao Autonomous Prefecture, Pingtang County, Zhangbu Town, Kala Village, 25°55′41.60″ N, 107°5′46.78″ E, 1000 m asl., 20 September 2019, C.Y. Deng (HGASMF01-2130). Hunan Province, Zhuzhou City, You County, Shangyunqiao Town, 27°2′20.68″ N, 113°21′34.67″ E, 90 m asl., 9 June 2017, Z.H. Chen (MHHNU 31049, epitype); Ningxiang City, Batang Town, Niaoming Village, 28°7′34.49″ N, 112°27′51.34″ E, 90 m asl., 20 July 2019, Z.H. Chen (MHHNU 31484).

Note. Molecular phylogenetical analysis showed that *R. brevispora* belongs to the subsect. *Pallidosporinae*, and has a relationship with *R. vesicatoria* Murrill from Costa Rica. However, *R. vesicatoria* can be easily distinguished by its unchanging context, larger basidiospores (7.3–7.9 × 6–6.7 µm), longer basidia (41–50 × 8–10.5 µm), and longer and more slender hyphal terminations (37–67 × 2–3 µm) [8,15]. In the field, *R. brevispora* was often misidentified as *R. japonica* owing to a white basidiomata, crowded lamellae, and stumpy stipe. However, our molecular phylogenetical results showed that *R. brevispora* is a new species distinguished from *R. japonica*. Morphologically, *R. japonica* differs with a longer stipe (50–60 mm in length), wider basidia (33–37 × 9.5–11.5 µm) and cylindric fusiform to long clavate hymenial cystidia [28].

#### 3.2.2. *Russula flavescens* (Y.L. Chen and J.F. Liang, sp. nov.)

MycoBank. MB 843373

Figure 2C,D, Figure 3C,D, Figure 6 and Figure 7

Diagnosis. *Russula flavescens* is similar to *R. japonica* and *R. pallidospora* J. Blum ex Romagn. However, it differs from *R. japonica* in its longer hymenial cystidia and from *R. pallidospora* in its longer basidiospores.

Etymology. “*flavescens*” means “becoming yellow” in Latin.

Holotype. CHINA, Jiangsu Province, Nanjing City, Zhongshan Botanical Garden, 32°3′31.56″ N, 118°49′53.81″ E, 65 m asl., 17 July 2021, Y.L. Chen (RITF5855), GenBank: ON010532 (ITS); ON751756 (nrLSU); ON751748 (mtSSU).

Basidiomata (Figure 2C,D) is small-to-medium-sized; pileus 35–80 mm in diameter; initially convex when young, then concave to infundibuliform when mature, obviously depressed at the center; surface smooth, dry, pure white, turning light yellow (4A5) to light brown (6D5–7D5) when touched or bruised, pellicle broken up into small patches; margin first inrolled, then expanded, rarely cracked, nonstriate. Lamellae adnate to slightly decurrent, 18–20 pieces at 1 cm near the pileus margin, white, staining light brown (7D5) or light orange (5A4) when bruised or touched; edge entire and concolor, lamellulae often present and irregular in length. Stipe central, 23–45 × 18–23 mm, subcylindrical, often tapering towards the base, smooth, white, staining yellowish (2A2) or grayish brown (6B6), solid. Context white, unchanging or changing grayish red (7B4) when bruised, compact; taste mild, odor slightly smelly. Spore print cream.

**Figure 6 jof-09-00061-f006:**
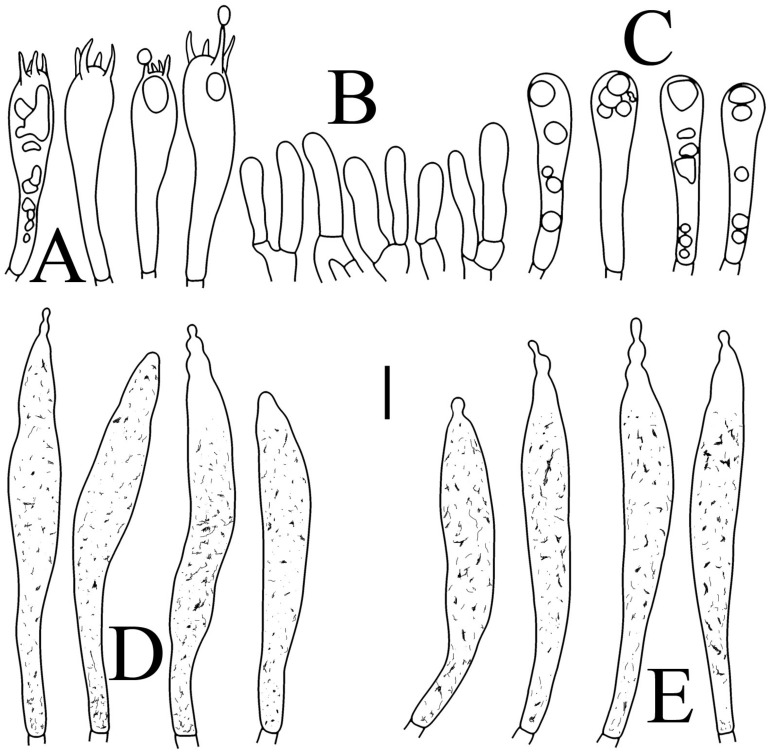
*Russula flavescens* (RITF5855, holotype): (**A**) basidia; (**B**) marginal cells; (**C**) basidiola; (**D**) hymenial cystidia on lamellae sides; (**E**) hymenial cystidia on lamellae edges. Scale bars = 10 μm.

**Figure 7 jof-09-00061-f007:**
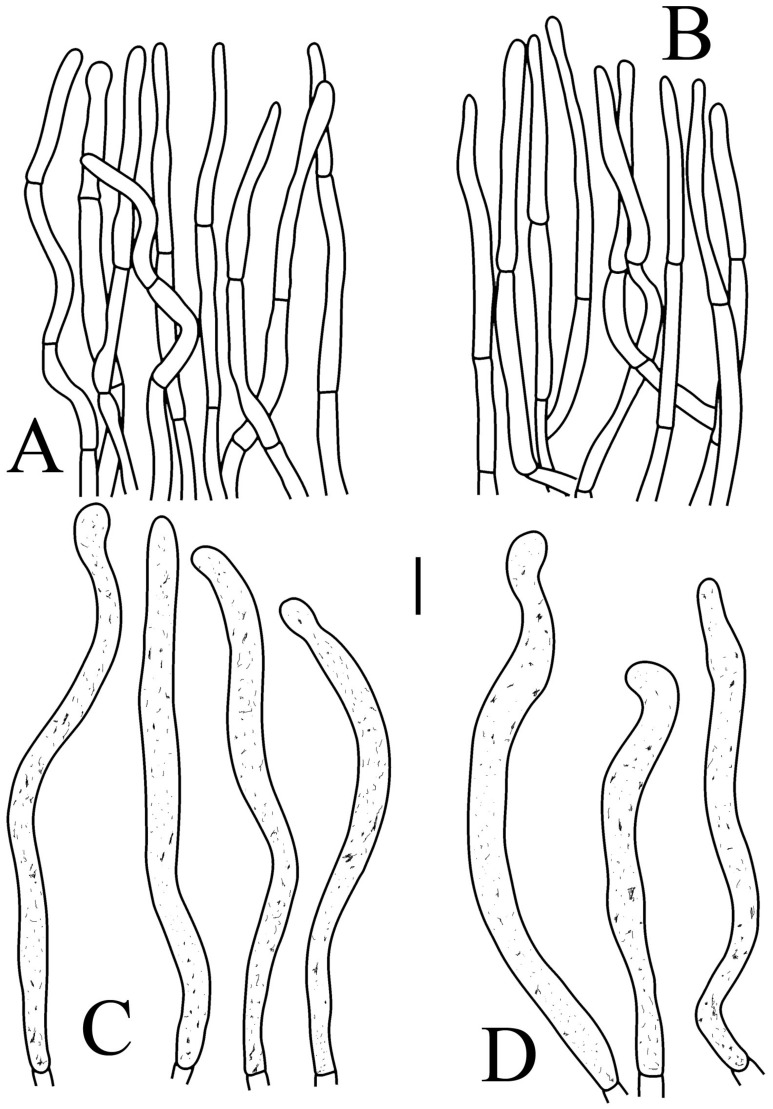
*Russula flavescens* (RITF5855, holotype): (**A**) hyphal terminations near the pileus margin; (**B**) hyphal terminations near the pileus center; (**C**) pileocystidia near the pileus margin; (**D**) pileocystidia near the pileus center. Scale bars = 10 μm.

Basidiospores (Figure 3C–D) (5.3–)5.6–5.8–6(–6.4) × (4.9–)5.1–5.4–5.6(–6) µm, Q = (1.01–)1.03–1.08–1.13(–1.19), globose to subglobose, sometimes broadly ellipsoid; ornamentation of small- to medium-sized, moderately distant to dense (5–7 in a 3 µm diameter circle) amyloid warts and spines, 0.6–0.9 µm high, subreticulate, rarely fused (0–1 in the circle), often connected by line connections [(0)1–4 in the circle]; suprahilar spot small-sized, inamyloid. Basidia (Figure 6A) (34.5–)36.5–39.5–42.5(–46) × (7–)8.1–8.8–9.5(–10) µm, 4-spored, rarely 2 or 3-spored, clavate; sterigmata up to 7.8 µm long; basidiola (Figure 6C) clavate, ca. 7.8–9.5 µm wide. Hymenial cystidia on lamellae sides (Figure 6D) widely dispersed to dispersed, ca. 200–500/mm^2^, (55.5–)61.5–69.5–78(–85) × (7.5–)8.5–9.5–10.5(–12.5) µm, fusiform, apically obtuse or rounded, occasionally acute, often with a submoniliform appendage, thin-walled; contents granulose or striated, no reaction in SV. Hymenial cystidia on lamellae edges (Figure 6E) dispersed, ca. 400–700/mm^2^, (55–)60–68.3–76.5(–88.5) × (8.4–)8.6–9.2–9.8(–11) µm, fusiform, apically obtuse, always with a papillary or submoniliform appendage, thin-walled; contents granulose, no reaction in SV. Marginal cells (Figure 6B) (10.5–)14.5–18–21.5(–23.5) × (3.5–)4.1–4.7–5.3(–5.5) µm, subclavate or cylindrical. Pileipellis orthochromatic in cresyl blue, not sharply delimited from the underlying context, single-layered, 186.8–383 µm deep, composed of dense, hyaline, horizontally oriented, 1.5–5.2 µm wide hyphae. Hyphal terminations near the pileus margin (Figure 7A) unbranched, thin-walled; terminal cells (21.2–)30–40.5–51.5(–59) × (2.9–)3–3.4–3.9(–4.6) µm, narrowly cylindrical, apically obtuse; subterminal cells cylindrical, ca. 1.8–6.1 µm wide, unbranched. Hyphal terminations near the pileus center (Figure 7B) slightly thinner than those near the pileus margin; terminal cells (28.6–)29.2–36–43(–46.5) × (2.7–)3–3.3–3.5(–3.6) µm, mainly cylindrical, apically obtuse; subterminal cells cylindrical, ca. 2.5–5.8 µm wide, unbranched. Pileocystidia near the pileus margin (Figure 7C) one-celled, (64.5–)80.5–93.5–106(–117) × (5.7–)5.9–6.5–7(–7.5) µm, clavate, apically obtuse, thin-walled, often flexuous; contents granulose, no reaction in SV. Pileocystidia near the pileus center (Figure 7D) one-celled, (75.5–)86.5–100–115(–126) × (4–)5.5–6.4–7.3(–8.1) µm, clavate, apically obtuse, thin-walled; contents granulose, no reaction in SV. Cystidioid hyphae frequent in context, with granulose contents; oleiferous hyphae frequent in context, with yellowish pigments.

Habitat. On the ground in broad-leaved forest of *Quercus variabilis* Blume.

Additional specimens examined. China, Jiangsu Province, Nanjing City, Zhongshan Botanical Garden, 32°3′31.39″N, 118°49′53.89″E, 70 m asl., 31 July 2021, Y.L. Chen (RITF5836); same place, 7 July 2022, W.J. Li (RITF6179). Yunnan Province, Chuxiong Autonomous Prefecture, Nanhua County, Yingwushan Park, 25°12′26.5″N, 101°15′29.61″E, 1930 m asl., 26 August 2018, F.C. Yang (RITF4454).

Notes. Phylogenetical analyses indicated that the new species belonged to subsect. *Pallidosporinae*. The ITS sequences of specimens MHHNU 31,366 (GenBank: OM760677) and MHHNU 31,881 (GenBank: OM760763) were submitted to the NCBI as *R. japonica* by Long [48]. However, close inspection of these collections indicated that they were different from *R. japonica*. *Russula japonica* can be easily distinguished by its shorter basidia (33–37 × 9.5–11.5 µm) and shorter hymenial cystidia (31–48 × 7.5–10.5 µm) [28]. Morphologically, *R. flavescens* also shares the same cream spore print, small basidiospores, and crowded lamellae with *R. brevispora*, *R. inopina* Shaffer, *R. pallidospora*, and *R. vesicatoria* in subsect. *Pallidosporinae*. However, *R. brevispora* has moderately numerous hymenial cystidia on lamellae sides and longer pileocystidia. *Russula inopina*, originally described from North America, is distinct in its longer and ellipsoid spores (6.9–7.5 × 5.2–5.7 µm), longer basidia (54–66 × 7.2–10 µm), and clavate hymenial cystidia [7,8]. *Russula pallidospora* differs in its longer stipe (40–45 × 18–28 mm), larger basidiospore (7.5–8.7 × 6–7 µm), longer basidia (48–62 × 8–11 µm), and hymenial cystidia (65–165 µm) [11,38], whereas *R. vesicatoria* is distinguished by its strong odor, bitter-acrid taste, larger basidiospores (7.3–7.9 × 6–6.7 µm), and moderately numerous hymenial cystidia [7,8].

#### 3.2.3. *Russula longicollis* (Y.L. Chen and J.F. Liang, sp. nov.)

MycoBank. MB 843374

Figure 2E,F, Figure 3E,F, Figure 8 and Figure 9

Diagnosis. *Russula longicollis* is distinguished from other known species in subsect. *Lactarioideae* by its mainly subglobose basidiospores (Q ≤ 1.15) and longer appendage of hymenial cystidia. It is similar to *R. leucocarpa* but differs in larger basidiospores.

Holotype. CHINA, Guangdong Province, Huizhou City, Boluo County, Luofu Mountain Provincial Nature Reserve, 23°16′5.64″ N, 114°3′24.92″ E, 100 m asl., 27 July 2021, Y.L. Chen (RITF5685), GenBank: ON010527 (ITS); ON751752 (nrLSU); ON745448 (*RPB2*); ON751744 (mtSSU).

Etymology. “*longicollis*” refers to the hymenial cystidia with a long neck.

Basidiomata (Figure 2E–F) medium- to large-sized; pileus 77–164 mm in diameter; initially convex to plano-applanate when young, then concave to shallowly infundibuliform, depressed at the center when mature; surface smooth, dry, white (1A1), staining yellowish-white (4A2), grayish-yellow (4B5) or grayish-orange (5B4) when bruised; margin rarely cracked, nonstriate. Lamellae adnate, 7–8 mm in height at the halfway, 12–15 pieces at 1 cm near the pileus margin, cream, staining pale yellow (4A3) when bruised; lamellulae frequently present and irregular in length, and furcations absent; edge entire and concolor. Stipe central, 30–42 × 16–30 mm, cylindrical, slightly narrow towards the base, smooth, white (1A1), staining pale yellow (4A3), solid. Context up to 10 mm thick at the pileus center, white (1A1), staining orange-white (5A2) when bruised, compact; taste slightly acrid, odor slightly putrid. Spore print cream.

Basidiospores (Figure 3E,F) (7.8–)8.1–8.5–8.6(–9.7) × (7.5–)7.7–8.0–8.3(–8.6) µm, Q = (1.01–)1.04–1.07–1.11(–1.15), mainly subglobose, sometimes globose; ornamentation of medium-sized, distant (3–6 in a 3 µm diameter circle), amyloid warts and ridges, up to 1.67 µm high, subreticulate, fused [(0)1–4(5) fusions in the circle], occasionally connected by fine line connections [(0)1–2 in the circle]; suprahilar spot medium-sized, amyloid. Basidia (Figure 8A) (47.5–)49.5–52.5–56(–58.5) × (10.2–)10.6–11.7–12.2(–12.9) µm, 2–4-spored, broadly clavate, colorless, thin to slightly thick-walled (1.09 µm); sterigmata up to 14.3 µm long; basidiola (Figure 8C) clavate, ca. 8.7–12.4 µm wide, colorless. Hymenial cystidia on lamellae sides (Figure 8E) dispersed to moderately numerous, ca. 600–900/mm^2^, (73–)77–83–89.5(–95.5) × (7.6–)8.5–9.1–9.7(–10) µm, fusiform or lageniform, apically obtuse, with a 6.5–14 µm long submoniliform and attenuate appendage, thin-walled, colorless; contents granulose-crystalline or banded, yellowish in SV. Hymenial cystidia on lamellae edges (Figure 8D) dispersed to moderately numerous, ca. 600–900/mm^2^, (61–)64–70.5–77(–84.5) × (7.8–)8.4–9.0–9.6(–10.4) µm, fusiform, apically obtuse, often with a 4.54–8.15 µm long submoniliform appendage, thin-walled, colorless; contents granulose or banded, yellowish-orange in SV. Marginal cells (Figure 8B) (13–)18.5–21.5–24.5(–26.5) × (5.8–)6.5–7.3–8.1(–8.8) µm, clavate or cylindrical, colorless. Pileipellis orthochromatic in cresyl blue, not sharply delimited from the underlying context, single-layered, 196–332 µm deep, composed of dense, horizontally oriented hyphae. Hyphal terminations near the pileus margin (Figure 9A) unbranched, rarely flexuous, thin-walled; terminal cells (21–)27–32.5–38(–41.5) × (3.1–)3.7–4.6–5.6(–7.1) µm, cylindrical or clavate, apically obtuse; subterminal cells cylindrical, ca. 2.6–5.3 µm wide, unbranched. Hyphal terminations near the pileus center (Figure 9B) thinner than those near the pileus margin; terminal cells (16–)20.5–26–31.5(–37.5) × (2.5–)3–3.5–4(–5.5) µm, mainly cylindrical, apically obtuse; subterminal cells cylindrical, ca. 2.5–4.2 µm wide, unbranched. Pileocystidia near the pileus margin (Figure 9C) always one-celled, (67–)75–90.5–109(–113) × (4–)4.5–5.5–6.5(–7.1) µm, subclavate or subcylindrical, apically obtuse, thin-walled, often flexuous; contents granulose, no reaction in SV. Pileocystidia near the pileus center (Figure 9D) always one-celled, (63.5–)69–80.5–91.5(–110) × (3.7–)3.9–4.7–5.5(–6.8) µm, subclavate or subcylindrical, apically obtuse, thin-walled; contents granulose, no reaction in SV. Cystidioid hyphae frequently present in context with granulose contents, and oleiferous hyphae in context present with yellowish pigments.

Habitat. On the ground in a broad-leaved forest dominated by *Castanopsis chinensis* Hance. or a mixed forest of *Quercus* sp. and *Pinus yunnanensis* Franch.

Additional specimens examined. CHINA, Yunnan Province, Dali Bai Autonomous Prefecture, Jianchuan County, 26°14′40.5″ N, 99°48′42.39″ E, 2190 m asl., 21 August 2020, J. Song (RITF5106); Binchuan County, 25°53′23.41″ N, 100°18′47.91″ E, 2450 m asl., 24 August 2022, X.L. Gao (RITF6388).

Notes. Phylogenetically, *R. longicollis* fell within subsect. *Lactarioideae*, and was close to *R. leucocarpa*. However, *R. leucocarpa* differs in its smaller basidiospores (5.4–6.7 × 4.5–5.9 µm), yellowish hymenial cystidia with granular contents, and habitat in coniferous forests [9]. Morphologically, *R. longicollis* resembles another three Chinese species *R. cremicolor* (G.J. Li and C.Y. Deng), *R. luteolamellata* (H. Zhou and C.L. Hou), and *R. subbrevipes* (J.F. Liang and J. Song). However, *R. cremicolor* can be distinguished by its subglobose to broadly ellipsoid (Q = 1.16 ± 0.07) basidiospores with completely reticulate ornamentation, basidia with shorter sterigmata (3−6 μm), and wider pileocystidia (6−8 μm) [9]. *Russula luteolamellata* (H. Zhou and C.L. Hou), a new species recently described from China easily differs in a yellowish to pale orange pileus, yellowish lamellae, and larger basidiospores (9–10.4 × 8–9.2 µm) [26]. *Russula subbrevipes* has a yellow ochre or yellowish-brown pileus and subglobose to broadly ellipsoid basidiospores [29]. Moreover, *R. longicollis* can be confused with other species within subsect. *Lactarioideae*. However, *R. brevipes* possesses subglobose to broadly ellipsoid basidiospores, larger basidia (55.5–68 × 9.5–14 µm), numerous hymenial cystidia, broader terminal hyphae, and weakly greying pileocystidia in SV [8]. *Russula chloroides* (Krombh.) Bres. differs from *R. longicollis* in a longer stipe (60−80 mm), greenish at the top of the stipe, longer basidiospores (8–11.2 × 7.2–8.8 µm), and larger hymenial cystidia with a mucronate to capitate appendage [10,11,38,49]. *Russula delica* Romagn. differs in its subdistant lamellae, longer basidiospores (8–11.5 × 6.5–8.7 µm) with lower ornamentation, longer hymenial cystidia (65–130 × 7.2–11.5 µm), and rare pileocystidia [7,10,11,38]. *Russula laevis* (Kälviäinen, Ruotsalainen and Taipale) has a smaller pileus (40–75 mm), slender stipe (12–20 mm), subglobose to broadly ellipsoid basidiospores, and longer hyphal terminations (36.5–71.5 × 5–7 µm) [2].

**Figure 8 jof-09-00061-f008:**
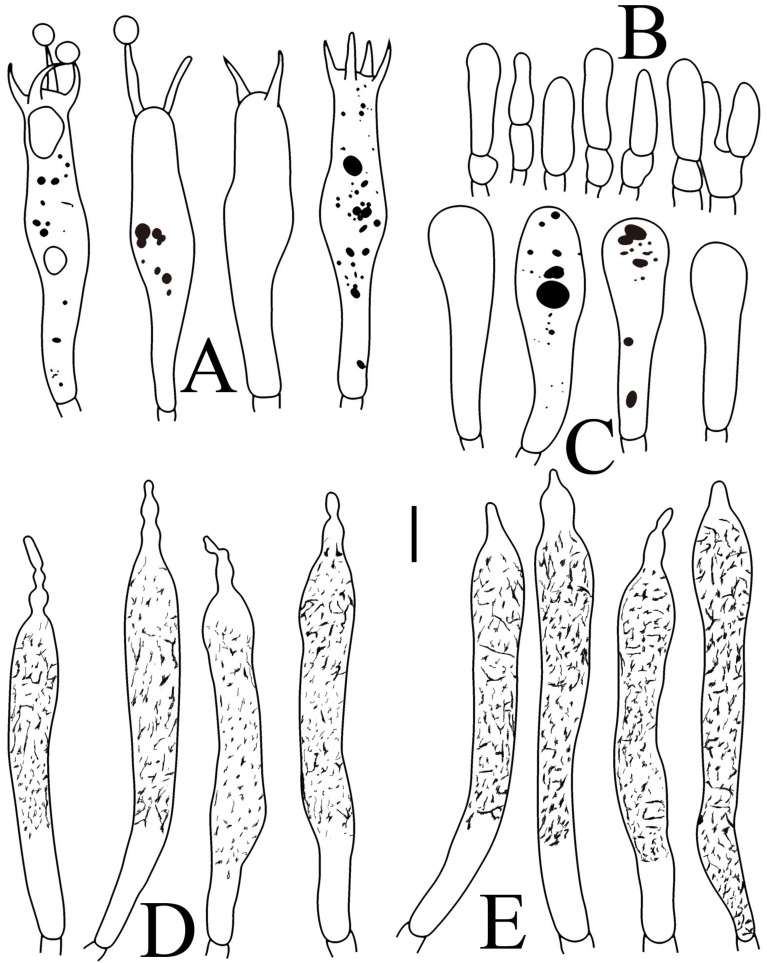
*Russula longicollis* (RITF5685, holotype): (**A**) basidia; (**B**) marginal cells; (**C**) basidiola; (**D**) hymenial cystidia on lamellae edges; (**E**) hymenial cystidia on lamellae sides. Scale bars: 10 μm.

**Figure 9 jof-09-00061-f009:**
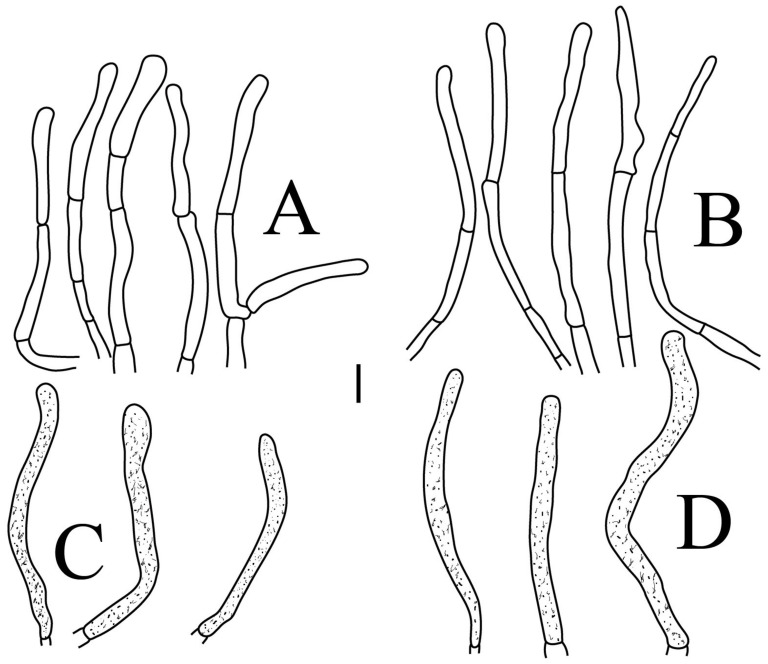
*Russula longicollis* (RITF5685, holotype): (**A**) hyphal terminations near the pileus margin; (**B**) hyphal terminations near the pileus center; (**C**) pileocystidia near the pileus margin; (**D**) pileocystidia near the pileus center. Scale bars: 10 μm.

#### 3.2.4. *Russula pseudojaponica* (Y.L. Chen and J.F. Liang, sp. Nov)

Mycobank. MB 843375

Figure 2G,H, Figure 3G,H, Figure 10 and Figure 11

Diagnosis. *Russula pseudojaponica* is similar to *R. japonica* but differs in its longer hymenial cystidia and basidia.

Holotype. CHINA, Sichuan Province, Zigong City, Fushun County, Qingshan Village, 29°5′4.16″ N, 105°6′17.82″ E, 420 m asl., 16 July 2021, Y.L. Chen & M.Y. An (RITF5755), GenBank: ON010533 (ITS); ON751757 (nrLSU); ON751747 (mtSSU).

Etymology. “*pseudojaponica*” refers to the morphological similarity to *R. japonica*.

Basidiomata (Figure 2G,H) medium-sized; pileus 35–70 mm in diameter; initially convex when young, then concave to infundibuliform when mature; surface smooth, dry, white (1A1) or cream when young, staining yellowish (3A2) or grayish-orange (5B4) at the center; margin rarely cracked, nonstriate, white (1A1) or cream. Lamellae adnate to slightly decurrent, 18–20 pieces at 1 cm near the pileus margin, white (1A1) to cream, unchanging when bruised; lamellulae often present and irregular in length, and furcations absent; edge entire and concolorous. Stipe central, 15–20 × 10–15 mm, cylindrical, often tapering towards the base, smooth, white (1A1), staining yellowish (2A2) or grayish-brown (6B6), solid. Context white (1A1), unchanging when bruised, compact; taste mild, odor slightly smelly. Spore print cream.

Basidiospores (Figure 3G–H) (5.9–)6.2–6.6–6.9(–7.5) × (5.6–)5.8–6.1–6.4(–7) µm, Q = (1.01–)1.04–1.08–1.12(–1.16), globose to subglobose; ornamentation of small to medium-sized, moderately distant to dense [5–8(9) in a 3 µm diameter circle] amyloid warts, low (less than 0.5 µm high), subreticulate, rarely fused (0–2 in the circle), often connected by fine line connections (0–1 in the circle); suprahilar spot small-sized, inamyloid. Basidia (Figure 10A) (31–)33.5–36.5–40(–42.5) × (7.4–)7.9–8.6–9.3(–9.9) µm, 2–4-spored, clavate, thin-walled; sterigmata up to 6.9 µm long; basidiola (Figure 10C) clavate, ca. 7.2–9.9 µm wide. Hymenial cystidia on lamellae sides (Figure 10D) dispersed to moderately numerous, ca. 600–900/mm^2^, (56.5–)66–75–84(–89.5) × (7.2–)7.8–8.9–10 (–11.7) µm, clavate or subfusiform, apically obtuse, often with a papillary appendage, thin-walled; contents granulose-crystalline, no reaction in SV. Hymenial cystidia on lamellae edges (Figure 10E) dispersed to moderately numerous, ca. 600–900/mm^2^, smaller than those on lamellae sides, (39–)44–52–60(–71.5) × (7.5–)7.8–8.2–8.6(–9) µm, clavate, subfusiform or sublageniform, apically rounded or obtuse, occasionally with a papillary or submoniliform appendage, thin-walled; contents granulose, sometimes banded, no reaction in SV. Marginal cells (Figure 10B) (10.5–)15–18.5–22.5(–26) × (4.1–)4.2–4.7–5.2(–5.8) µm, subclavate or cylindrical. Pileipellis orthochromatic in cresyl blue, sharply delimited from the underlying context, two-layered, composed of suprapellis and subpellis; suprapellis a trichoderm, 160–267 µm deep, composed of thin-walled, narrowly cylindrical, colorless hyphae; subpellis a cutis, 131–310 µm deep, composed of gelatinized, interweaved colorless hyphae 2.5–5.4 µm wide, with sphaerocytes 12.8–39.9 µm wide. Hyphal terminations near the pileus margin (Figure 11A) unbranched, rarely flexuous, thin-walled; terminal cells (20.5–)24.5–33–41(–53) × (3–)3.2–3.5–3.9(–4.3), cylindrical or clavate, apically obtuse; subterminal cells cylindrical, ca. 2.6–4.5 µm wide, unbranched. Hyphal terminations near the pileus center (Figure 11B) similar to those near the pileus margin; terminal cells (20–)21–28.5–35.5(–43.5) × (2.6–)2.9–3.4–3.9(–4.5) µm, mainly cylindrical, apically obtuse; subterminal cells cylindrical, ca. 3–4.7 µm wide, unbranched. Pileocystidia near the pileus margin (Figure 11C) always one-celled, 51.5–70–109(–128) × (4.5–)5–6.2–7 (–8) µm, mainly cylindrical, apically obtuse, sometimes with a papillary appendage, thin-walled; contents granulose, no reaction in SV. Pileocystidia near the pileus center (Figure 11D) always one-celled, (50–)58–73–88(–106) × (5–)5.6–6.4–7.2(–7.7) µm, mainly cylindrical, apically obtuse, sometimes with a papillary appendage, thin-walled; contents granulose or banded, no reaction in SV. Cystidioid hyphae frequent in subpellis and context with granulose contents; oleiferous hyphae frequent in subpellis and context with yellowish pigments.

Habitat. On the ground in a mixed forest of *P. massoniana* and *Quercus* sp.

Additional specimens examined. CHINA, Guizhou Province, Tongren City, Fanjingshan National Nature Reserve, 27°54′43.25″ N, 108°39′25.62″ E, 1950 m asl., 1 August 2019, C.Y. Deng (HGASMF-009923).

Notes. Phylogenetically, *R. pseudojaponica* falls into subsect. *Pallidosporinae*, and is closely related to *R. flavescens*, *R. japonica*, *R. brevispora*, and *R. vesicatoria*. However, *R. flavescens* differs in its pure white basidiomata, which turn light yellow to light brown when touched or bruised, slightly larger basidiospores, and clavate hymenial cystidia on lamellae edges. *Russula japonica* is distinct from *R. pseudojaponica* in its larger fruiting bodies (60–140 mm), wider basidia (33–37 × 9.5–11.5 µm), and shorter hymenial cystidia (31–48 × 7.5–10.5 µm) [28]. *Russula brevispora* differs in lamellae and context staining grayish-orange, shorter hymenial cystidia and pileocystidia, and two forms of hyphal terminations in the pileipellis. *Russula vesicatoria* differs in its incurved pileus margin when mature, strong and pleasant odor, and astringent to bitter taste rapidly becoming very acrid [7,8]. Morphologically, *R. pseudojaponica* is also easily confused with *R. littoralis* Romagn., *R. flavispora* Romagn., *R. pallidospora*, and *R. pseudodelica* J.E. Lange. *Russula littoralis*, originally described from the Central America, differs in its longer basidia (46–56 × 8.5–10 µm) and its dispersed and slender hymenial cystidia (70–80 × 5–8.5 µm) [38]. The European *R. flavispora* differs in its larger pileus (60–120 mm), larger basidia (45–60 × 9.5–11 µm), longer cystidia (80–115 × 6–9 µm), and yellow spore print [11,38,49]. *Russula pallidospora*, originally described from Europe, has larger basidiospores (7.5–8.7 × 6–7 µm), longer basidia (48–62 × 8–11 µm), and fusiform and longer hymenial cystidia (65–160 µm) [11,38]. *Russula pseudodelica* differs in its larger stipe (30–60 × 20–30 mm), sparsely aculeate basidiospores, and absent pileocystidia [10,11].

**Figure 10 jof-09-00061-f010:**
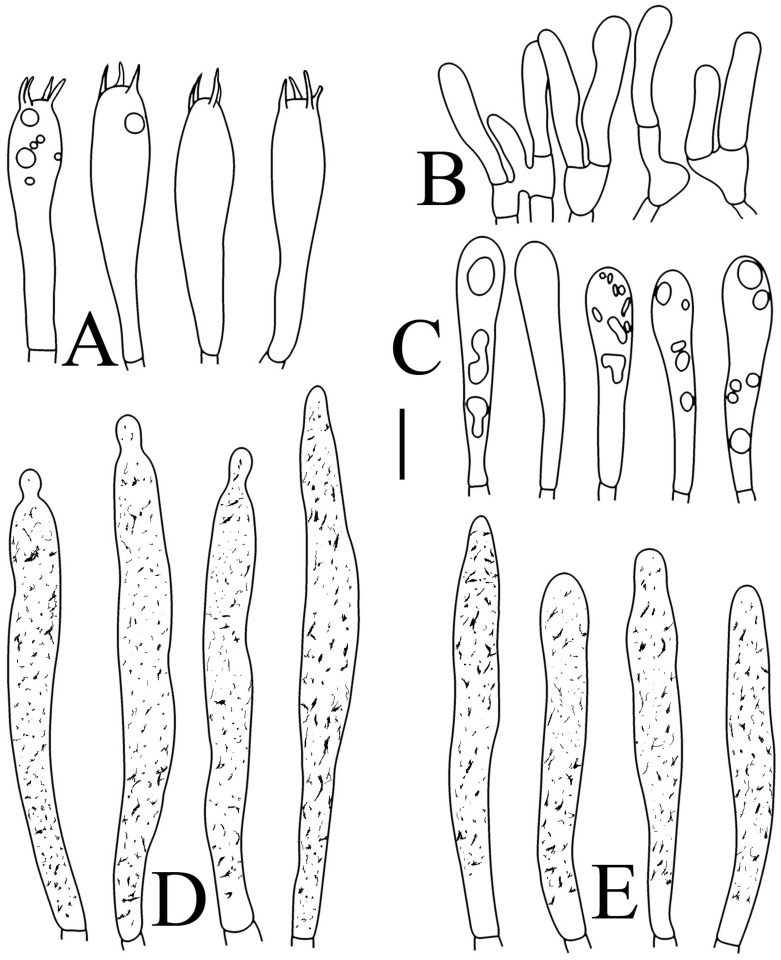
*Russula pseudojaponica* (RITF5755, holotype): (**A**) basidia; (**B**) marginal cells; (**C**) basidiola; (**D**) hymenial cystidia on lamellae sides; (**E**) hymenial cystidia on lamellae edges. Scale bars: 10 μm.

**Figure 11 jof-09-00061-f011:**
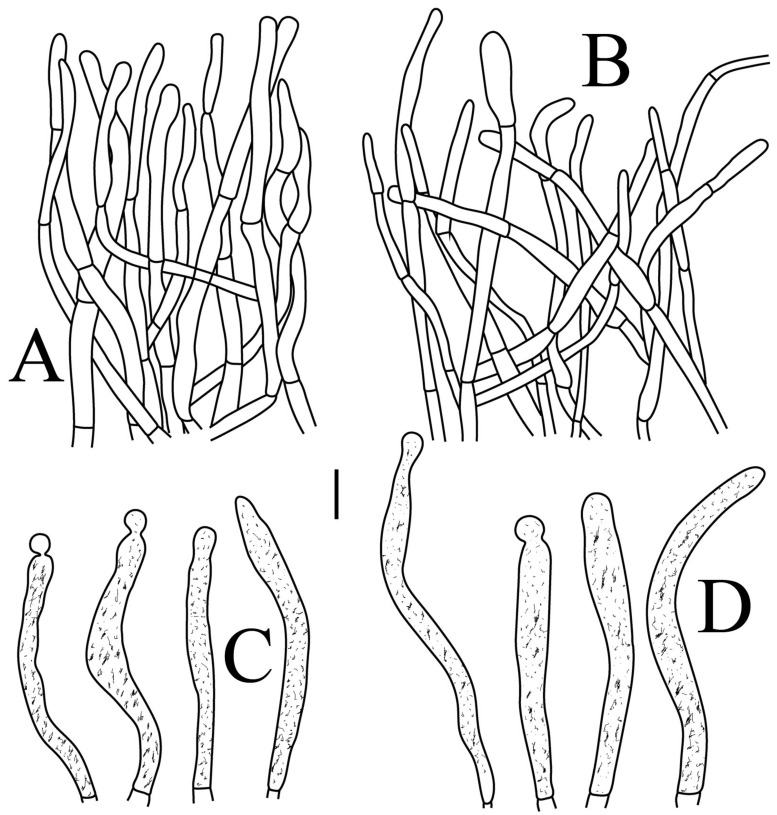
*Russula pseudojaponica* (RITF5755, holotype): (**A**) hyphal terminations near the pileus margin; (**B**) hyphal terminations near the pileus center; (**C**) pileocystidia near the pileus margin; (**D**) pileocystidia near the pileus center. Scale bars: 10 μm.

## 4. Discussion

*Russula* subg. *Brevipedum* were formerly placed in *R.* subg. *Compactae* (Fr.) Bon as section *Brevipes* and elevated to subgenus in 2015 [5]. In China, 13 species of subg. *Brevipedum* were reported in the past, with six species—*R. byssina* (G.J. Li and C.Y. Deng), *R. callainomarginis* (J.F. Liang and J. Song), *R. cremicolor*, *R. leucocarpa*, *R. luteolamellata*, and *R. subbrevipes*—described as new in recent years [9,25,26]. The other species (*R. chloroides*, *R. delica*, *R. pallidospora*, *R. pseudodelica*, *R. flavispora*, *R. japonica*, and *R. brevipes*) were described in detail in various publications [24,27,30]. The species described in China were similar to those of Europe and North America in terms of macroscopical morphology, but there were obvious differences in microscopic characteristics. Moreover, most of the species reported in China lack support from molecular evidence.

*Russula japonica* was first described by Hongo [28], and was reported to be poisonous, causing gastroenteritis [33,34,35]. Over the past decades, most of the specimens with similar characteristics were recorded as *R. japonica* in China. Our morphological study results indicate that the Chinese species differ from *R. japonica* in their hymenial cystidia. Moreover, we selected a large ribosomal submit sequence from the location of type provided by Shimono [50] for molecular phylogenetical analyses. The molecular phylogeny results are consistent with the morphology.

In some other species, the differences between Chinese and North American and European specimens mainly appear in terms of the size of the basidiospores, hymenial cystidia, and basidia, the presence or absence of pileocystidia, and so on. For example, *R. chloroides* has longer basidiospores (7.5–8.9 × 5–6.7 μm in Li [9] and 8.3–11.4 × 7.7–10.3 μm in Liu [24] and shorter hymenial cystidia (34–67 μm in Li [9] and 55.3–91.1 μm in Liu [24]), but smaller basidiospores (6.5–8 × 6–6.7 μm) and longer hymenial cystidia (65–115 μm) are present in Romagnesi [30]. *Russula brevipes* has shorter basidia (17–46 × 5–12 μm in Li [9] and 37.8–48.2 × 11.8–13.6 μm in Liu [24]), shorter hymenial cystidia (29–64 × 5–12 μm in Li [9] and 56.3–81.5 × 6.9–12.6 μm in Liu [24]), and the absence of pileocystidia; however, there are longer basidia (55.5–68 × 9.5–14 μm) and hymenial cystidia (62–93 × 7.5–11 μm), along with the presence of pileocystidia, weakly greying in SV, in Buyck [8]. This indicates that these species reported from China might be misidentified. The main reason for this is that most descriptions of *Russula* species in the past have been incomplete. Besides being incomplete, the description style has not been consistent, with regional or author-specific patterns [2]. The combination of these factors makes comparing descriptions difficult, or even impossible. Therefore, we still need more reliable sequences as well as detailed macroscopic and microscopic descriptions to confirm the existence of these species in China.

In this study, we propose four new species, including three species that are closely related to *R. japonica* and one species that is close to *R. leucocarpa* from China based on molecular phylogeny and morphology data. It is worth noting that *R. flavescens*, *R. pseudojaponica,* and *R. brevispora* were recently reported to have gastrointestinal toxicity [34,35], whereas the toxicity of *R. longicollis* is still unknown.


**Key to four new species and other closely related taxa within subg. *Brevipedum***


1 Basidiospores with inamyloid suprahilar spot………………………………..……….…….2

1 Basidiospores with amyloid suprahilar spot………………………………..………….…..9

2 Spore print yellow………………………………………….………………………*R. flavispora*

2 Spore print white to cream…………………………………………………….………..….…3

2 Spore print ochraceous………………………………………………………….………….….8

3 Hymenial cystidia on lamellae edges usually <50 µm in length……………….*R. japonica*

3 Hymenial cystidia on lamellae edges usually >50 µm in length………………..…….….4

4 Basidiospores on average < 7 µm in length……………………………………..……………5

4 Basidiospores on average > 7 µm in length……………………………….……….…..…….7

5 Lamellae unchanging when bruised………………………………………*R. pseudojaponica*

5 Lamellae turning yellowish or brownish when bruised……………………………………6

6 Moderately numerous hymenial cystidia on lamellae sides, 42.7–60 × 8.4–10.2 µm, no reaction in SV; comparatively longer pileocystidia……………………………..*R. brevispora*

6 Widely dispersed to dispersed hymenial cystidia on lamellae sides, 61.5–78 × 8.5–10.5 µm, grayish in SV; comparatively shorter pileocystidia………………..……….*R. flavescens*

7 Hymenial cystidia comparatively short, 70–80 µm……………………………..*.R. littoralis*

7 Hymenial cystidia comparatively long, 65–160 µm…………………………*R. pallidospora*

8 Basidiospores comparatively short, 7.3–7.9 × 6–6.7 µm; Basidia comparatively small, 41–50 × 8–10.5 µm………….…………………………………….……………….…….*R. vesicatoria*

8 Basidiospores comparatively long, 6.9–9.3 × 6.3–7 µm; Basidia comparatively large, 48–62 × 9–11 µm…………………………………………..………………..………….*R. pseudodelica*

9 Pileus surface yellow-ochre or yellowish-brown……………………………….………….10

9 Pileus surface white to cream………………………………………………….……..….….11

10 Basidiospores comparatively large, 9–10.4 × 8–9.2 µm…………………..*R. luteolamellata*

10 Basidiospores comparatively small, 7.8–9 × 6.9–7.9 µm……………..……..*R. subbrevipes*

11 Basidiospores on average < 7 µm in length…….……………………………..*R. leucocarpa*

11 Basidiospores on average > 7 µm in length…………………………………….….……..12

12 Lamellae comparatively distant (usually less than 12 pieces/cm)……….………*R. delica*

12 Lamellae comparatively crowded (usually more than 12 pieces/cm)……………..…..13

13 Basidiospores globose to subglobose…………………………………….….…*R. longicollis*

13 Basidiospores subglobose to broadly ellipsoid………………………….……..……..….14

14 Basidiospores on average < 10 µm in length……………………………….……*R. brevipes*

14 Basidiospores on average > 10 µm in length……………………………..…….…….….15

15 Stipe comparatively broad (20–30 mm); hymenial cystidia comparatively wide (7–13 µm), more ellipsoid terminal cells in the pileipellis………………………………*R. chloroides*

15 Stipe comparatively slender (12–20 mm); hymenial cystidia comparatively narrow (7.5–8.5 µm), more clavate terminal cells in the pileipellis…………………………………*R. laevis*

## Figures and Tables

**Table 1 jof-09-00061-t001:** Taxa, vouchers, and GenBank accession numbers of sequences analyzed in this study.

Taxa	Vouchers	Location	NCBI. No			
			ITS	nrLSU	*RPB2*	mtSSU
*R. allochroa*	JAC13329	New Zealand	MW683836	MW683672		
	JAC16279	New Zealand	MW683883	MW683710		
*R.* cf. *angustispora*	PC BB2004-252	USA	EU598152			
*R. aucarum*	PC0124732	Unknown	KY800351			
*R. australis*	JAC10732	New Zealand		MW683616		
*R. brevipes*	226/BB 06.508	Mexico		KU237479	KU237765	KU237323
	BPL707	USA	KY848511			
	JS160927-01	South Korea	MG407682			
	SMI329	Canada	FJ845429			
*R.* cf. *brevipes*	232/BB 06.441 *	Mexico		KU237483	KU237769	KU237327
** *R. brevispora* **	**HGASMF-009915 ***	**China**	**MN648956**	**OP850852**		**OP856542**
	**MHHNU 31049**	**China**	**MK167414**			**OP856544**
	**MHHNU 31484**	**China**	**OM760732**			**OP856541**
	**HGASMF01-2130**	**China**	**OP850833**			**OP856541**
*R. cascadensis*	UBC F19691	Canada	HM240541			
	UBC:F30189	Canada	KX812838			
*R. byssina*	HGASMF-009907 *	China	MN648951			
*R. callainomarginis*	RITF2639 *	China	MH286463	MH286468	MH911624	MH911616
*R.* aff. *chloroides*	FH12273	Belgium		KT933876	KT933947	
*R. chloroides*	205RUS24	Europe	AY061663			
	572/BB 07.209	Slovakia		KU237559	KU237845	KU237407
	HA1	Iran	KU237246			
	IZS73	Italy	MZ005492			
	TUB ue68	Germany	AF418604			
*R. cremicolor*	HGASMF-009901 *	China	MN648955			
	**HGASMF01-4368**	**China**	**OP693466**	**OP850856**		**OP856547**
	**HGASMF01-8543**	**China**	**OP693465**	**OP850855**	**OP856856**	**OP856546**
	**HGASMF01-3661**	**China**	**OP693464**	**OP850854**	**OP856855**	**OP856548**
	**HGASMF01-4362**	**China**	**OP850832**			
*R.* aff. *delica*	1119/BB 12.086	Italy		KU237594	KU237879	KU237442
*R. delica*	496RUS26	Europe	AY061671			
	FH 12-272	Belgium	KF432955	KR364224	KR364340	
	H21527	Tunisia	KU973863			
	HMJAU 32182	China	KX094989			
	KA12-1327	Korea	KR673555			
*R.* cf. *delica*	1195/SA07.210	Slovakia		KU237600	KU237885	KU237449
*R. dissimulans*	BPL285	USA	KT933979	KT933840	KT933911	
** *R. flavescens* **	**RITF4544**	**China**	**ON010530**			
	**RITF5836**	**China**	**ON010531**	**ON751758**	**ON745449**	**ON751749**
	**RITF5855 ***	**China**	**ON010532**	**ON751756**		**ON751748**
	**RITF6179**	**China**	**OP850835**			
	MHHNU 31881	China	OM760763			
	MHHNU 31366	China	OM760677			
*R. fuliginosa*	FH RUS 14091001	Slovakia	MW172330	MW182487	MW306693	
*R. herrerae*	239/BB 06.532	Mexico		KU237486	KU237772	KU237330
*R. japonica*	OSA-MY-1709	Japan		AB154697		
*R. laevis*	CLC_1146	USA	MT583330		MT500691	
	CLC_1883	USA	MT583337		MT500692	
	KUO (JR4016)	Finland		MN130128	MN380529	MN161180
	KUO (JR4376)	Finland		MN130129		MN161181
*R. leucocarpa*	HGASMF-009910 *	China	MN648948			
	**HGASMF01-6534**	**China**	**OP850834**	**OP850853**		**OP856549**
*R. littoralis*	1222IS87	Europe	AY061702			
** *R. longicollis* **	**RITF5106**	**China**	**ON010526**	**ON751753**		**ON751743**
	**RITF5685 ***	**China**	**ON010527**	**ON751752**	**ON745448**	**ON751744**
	**RITF6388**	**China**	**OP693462**			
*R. marangania*	MEL2293694	Australia	EU019930	EU019930		
*R. nigrocarpa*	K18050529	China	MN688795			
*R. pallidospora*	2-1221IS85	Europe	AY061701			
*R. pseudojaponica*	HGASMF-009923	China	MN648957		**OP856854**	**OP856544**
	**RITF5755 ***	**China**	**ON010533**	**ON751757**		**ON751747**
*R. pumicoidea*	Trappe 14771	Australia	EU019931	EU019931		
*R. sinuata*	H4755*	Australia	EU019943			
*R.* sp.	233/BB 06.476	Mexico		KU237484	KU237770	KU237328
***R.* sp.**	**RITF3420**	**China**	**ON010528**	**ON751754**	**ON745450**	**ON751745**
	**RITF3421**	**China**	**ON010529**	**ON751755**		**ON751746**
*R. subbrevipes*	RITF2946	China	MH286462	MH286467		MH911618
	RITF3002	China	MH286461	MH286466		MH911619
	RITF3136 *	China	MH286460	MH286465	MH911625	MH911617
	**RITF6148**	**China**	**OP693466**			
*R. subnigricans*	ZP7036 MHHNU	China	EF126734			
*R. vesicatoria*	1187/BB 07.034	USA		KU237599	KU237884	KU237447
	PC0124666	Costa Rica	KY800359			

Type specimens are marked with an asterisk (*) symbol, and the newly generated sequences are in bold.

## Data Availability

Not applicable.

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
