# Peer review of "Morphological Characteristics and Molecular Evidence Reveal four New Species of Russula subg. Brevipedum from China"

_jof, 2022, doi:10.3390/jof9010061_

Round 1
Reviewer 1 Report (Previous Reviewer 3)
Minor observations are sent in the manuscript

Author Response
Dear reviewers,
Thank you for your valuable comments on my article. I have improved my article as follows:
- in line 223, corrected “Pallidosporinae”.
- In line 243, 329, 413, without italics.
- In line 246, “Pallidosporinae” and line 510 “R. flavescens”
- “much lageniform hymenial cystidia” was changed to “lageniform hymenial cystidia”.
- In line 309, “Molecular phylogenetical analysis, japonicoidea belongs to the subsect. Pallidosporinae, and has some relationship with R. vesicatoria which was from Costa Rica” was changed to “Molecular phylogenetical analysis showed that R. brevispora belongs to the subsect. Pallidosporinae, and has some relationship with R. vesicatoria Murrill from Costa Rica.”
- In line 397, delate “and”.
- In line 471, “ flavescens differs in its unchanging pileus and lamellae” was changed to “R. flavescens differs in its pure white basidiomata, which turn light yellow to light brown when touched or bruised”.

Reviewer 2 Report (Previous Reviewer 1)
I’m very glad to review this paper about taxonomy, phylogeny of Russula subg. Brevipedum and the submission is worthy of publication after carefully revising. My detailed comments are as follows:
1. The language used in this article has to be carefully examined and polished, with particular attention to the following three points:
(1) Accurate use and spelling of vocabularies is needed here (see notes in appendix for details);
(2) The font and abbreviation of the Latin name should be unified, for example, “Russula” could be unified as “R.” after the previous mention;
(3) Some sentences may have grammar problems, and the sentence expression is single, so it is necessary to check and polish the expression (some inappropriate sentences have been highlighted in the attachment).
2. In “Materials and Methods”, followed information should be added: (1) the methods and software used for sequences combination; (2) a submission to Treebase of alignment for phylogenetic studies.
3. In Taxonomy, a supplement of key for species of Russula subg. Brevipedum or at least species mentioned in this paper is better.
4. Diagnosis of your proposed new species should be focus on the characteristics (a refined description) of itself rather than a comparison of resemble species.
5. In Figure 3, definition of image in A, D, and H is not enough, and a further adjustment is necessary here.

Author Response
Dear reviewer,
First of all, thank you very much for your comments. We have improved the paper according to these valuable emendations. The specific modifications are as follows:
- In the title, “characters” was changed to “characteristics”.
- In line 28, “Lactarioideae”,
- In line 33, “is one of the most species genera” was changed to “is a genera with great diversity”
- In 42-47, the sentence “The members of this subgenus mostly have medium to very large basidiomata that are thick-fleshed, a whitish pileus and stipe often rapidly developing yellowish brown to reddish brown stains, regularly unequal lamellae, white to cream context turning yellowish to rusty brown mostly with distinct smell or acrid to strongly acrid taste, whitish to yellow spore print, mucronate to obtuse-rounded cystidia in all parts of the fruiting body” was changed to “The members of this subgenus mostly have a medium to very large basidiomata, which often stains yellowish-brown to reddish-brown, regularly unequal lamellae, a distinct smell or acrid to strongly acrid taste, a whitish to yellow spore print, and mucronate to obtuse-rounded cystidia in all parts of the fruiting body”.
- In line 88, “The DNA was amplified with the primers: ITS1 and ITS4 for ITS” was changed to “The ITS region of rDNA was amplified with the primers ITS1 and ITS4 ”
- In line 91, “nrLSU, RPB2, and mtSSU were amplified using the primers and protocols comply with” was changed to “The remaining three loci (nrLSU, RPB2, and mtSSU) were amplified using the primers and protocols complying with Buyck”.
- The method of combining sequences should be mentioned here.
A: The method of combining sequences has been added in the revision. “The four datasets were concatenated using Phyutility v2.2 for further analysis” in line 104-105.
- A submission to Treebase of alignment for phylogenetic studies.
A: The final aligned result has been submitted to TreeBase (S29992). A submission result have been indicated in line 105-106.
- In line 472, “lightly” was corrected to “slightly”.
- In line 510, “R. flavescens”
- In Taxonomy, a supplement of key for species of Russula Brevipedum or at least species mentioned in this paper is better.
A: In the revised version, key to four new species and other closely related taxa within subg. Brevipedum has been added.
- Diagnosis of your proposed new species should be focus on the characteristics (a refined description) of itself rather than a comparison of resemble species.
A: The diagnosis description of our paper is based on the latest standards for new species proposed by Aime (2021, 10.1186/s43008-021-00063-1)
- In Figure 3, definition of image in A, D, and H is not enough, and a further adjustment is necessary here.
A: We have changed a picture with higher definition.
- The font and abbreviation of the Latin name should be unified, for example, “Russula” could be unified as “R.” after the previous mention;
A:The generic name should not be abbreviated at the beginning of a sentence.

Reviewer 3 Report (New Reviewer)
The paper submitted to my evaluation assesses an interesting and notoriously confused group of Russula, i.e. white-coloured species of the R. delica-complex. The aim of the article is to introduce three new species found in China, improving the knowledge of this group already documented by several recent publications.
I note that the paper is well-formatted and with apparently adequate illustrations of the new species; the microscopical descriptions sound accurate and follow the current standards of descriptions of Russulas. The phylogenetic analysis is not esthetically refined but sufficiently convincing for supporting the distinction of the four newly proposed species.
I have several comments to make on the paper :
1) First and the most critical, the English language needs a complete revision. Almost no sentence in the text is correctly written : grammatical rules and appropriate vocabulary are ignored, except in descriptions which are technically and linguistically correct.
2) Bibliographic data are partly incomplete and most of them need re-formatting (capitals on words of titles etc.). For instance :
6. Peck... 1890. Pages missing
11. Bon M. Monographic key for European Russulae. Mycologic Document 1988, 71-72, 1-125 is actually : Bon M. Clé monographique des Russules d'Europe. Docum. mycol. 18(70-71): 1-125 (original title in French)
14. Buyck B. 2003 : does not concern a species of subgen. Brevipedum but R. cremeolilacina. The same confusion seems to be repeated p. 18 line 481 (if the authors read the description of R. cremeolilacina they will not find any common feature between this species and a Brevipedum). I suspect an odd confusion between Russula littoralis Pegler (an illegitimate name put in synonymy with R. cremeolilacina by Buyck) and Russula littoralis Romagn. which is a real Pallidosporinae figuring in the tree of fig. 1.
15. Barbosa 2016. Editor and pages missing.
18. Cooper & Leonard 2014. Editor and pages missing.
I find useless the inflation of reference citations due to Table 1 (refs 45 to 59), since all are directly accessible through the GenBank Accession Number cited in the table.
3) l. 134, l. 509, and fig. 1 a "R. lactarioideae" is mentioned as unpublished species (as "holotype" in fig. 1 !), and in tab. 1 and fig. 1 a "R. delicinae" is mentioned. These names do not exist as epithets (valid or invalid), but as sections or subsections. Anyway, they do not correspond to the name attached to the sequences in GenBank :
KU237483 ("R.lactarioideae") is "Russula aff. brevipes"
KU273484 ("R. delicinae") is "Russula sp."
I could not trace an source for these differences, I suspect that they were taken from an already erroneous dataset published elsewhere.
4) Table 1 needs to be checked carefully. The bibliographic references cited in GenBank for some sequences do not match those cited for the same sequences in the table. (for instance KU237483, KU273484, etc.). The location is not standardized (sometimes a region, a country, a continent... "Filand" is to be corrected as "Finland".
5) In Fig. 1 many even minor differences are perceptible in such groups of sequences as "R. flavescens". I suppose that this comes from low-quality sequences of an insufficient check of chromatograms for these new sequences.
6) Fig. 1 again, long-branch anomalies are visible for the 2 "R.laevis KUO" sequences from Finland, and for "R. delica H21527 Tunisia", which are only explicable by a problem in the alignment or when concatenating the sequences for the multigene analysis. It should be checked and resolved from the alignment.
7) line 171 : "flavescens" does not mean "turning yellowish bodies when bruised", but just "becoming yellow" in Latin.
8) Line 306 : "MHHNU 31049, epitype" for R.japonicoidea : probably coming from an earlier version of the manuscript, no epitype to be designated for this new species.
9) Fig. 1 again, "subsect. Pallidosporinae" instead of "subsect. Pallidosporinea"
In conclusion, this paper deserves a careful revision (I am pretty sure that a number of details escaped to my attention, but a really meticulous revision of the data would have pushed this review out of delays). This concerns especially Table 1 (with elimination of the superfluous bibliography), Figure 1 which I suggest to reload from an improved alignment, and the most striking mistakes I found in the text. A special effort must be put on the English language all along the text.
The description of new species undoubtedly well-supported by molecular data and careful descriptions is to be supported anyway, and I warmly encourage the authors to improve the manuscript and submit a revised version to this journal. I would be glad to evaluate it again.
Author Response
Dear reviewer,
First of all, thank you very much for your valuable and specific comments on my paper. We have made changes to these comments one by one, and the specific changes and explanations are as follows:
1) First and the most critical, the English language needs a complete revision. Almost no sentence in the text is correctly written: grammatical rules and appropriate vocabulary are ignored, except in descriptions which are technically and linguistically correct.
A: We have improved the use of grammar and vocabulary.
2) Bibliographic data are partly incomplete and most of them need re-formatting (capitals on words of titles etc.). For instance :
- Peck... 1890. Pages missing
A: As the sentence of the article has been changed, this reference has been deleted.
- Bon M. Monographic key for European Russulae. Mycologic Document 1988, 71-72, 1-125 is actually : Bon M. Clé monographique des Russules d'Europe. Docum. mycol. 18(70-71): 1-125 (original title in French)
A: This reference has been perfected as “Bon, M. Clé monographique des Russules d'Europe. Docum. mycol. 1988, 18 (70-71), 1–125.”
- Buyck B. 2003 : does not concern a species of subgen. Brevipedum but R. cremeolilacina. The same confusion seems to be repeated p. 18 line 481 (if the authors read the description of R. cremeolilacina they will not find any common feature between this species and a Brevipedum). I suspect an odd confusion between Russula littoralis Pegler (an illegitimate name put in synonymy with R. cremeolilacina by Buyck) and Russula littoralis Romagn. which is a real Pallidosporinae figuring in the tree of fig. 1.
A: This reference has been deleted.
- Barbosa 2016. Editor and pages missing.
A: This reference has been perfected as “Duque Barbosa, J.A. Análise filogenética de Russula pers. (Russulaceae, Russulales: Agaricomy-cetes). Thesis of Master, Federal University of Santa Catarina, Florianopolis, 2016, 1–94.”
- Cooper & Leonard 2014. Editor and pages missing.
A: This reference has replaced to “McNabb, R.F.R. Russulaceae of New Zealand 2. Russula Pers. ex S. F. Gray, New Zeal. J. Bot. 2003, 11, 673–730.”
I find useless the inflation of reference citations due to Table 1 (refs 45 to 59), since all are directly accessible through the GenBank Accession Number cited in the table.
A: We have deleted the list of references in Table 1.
3) l. 134, l. 509, and fig. 1 a "R. lactarioideae" is mentioned as unpublished species (as "holotype" in fig. 1 !), and in tab. 1 and fig. 1 a "R. delicinae" is mentioned. These names do not exist as epithets (valid or invalid), but as sections or subsections. Anyway, they do not correspond to the name attached to the sequences in GenBank :
KU237483 ("R.lactarioideae") is "Russula aff. brevipes"
KU273484 ("R. delicinae ") is "Russula sp."
A: We have changed “R. lactarioideae” to “Russula aff. brevipes” and “R. delicinae” to "Russula sp." In Table 1 and Figure 1.
I could not trace an source for these differences, I suspect that they were taken from an already erroneous dataset published elsewhere.
4) Table 1 needs to be checked carefully. The bibliographic references cited in GenBank for some sequences do not match those cited for the same sequences in the table. (for instance KU237483, KU273484, etc.). The location is not standardized (sometimes a region, a country, a continent... "Filand" is to be corrected as "Finland".
A: We have unified the location information of the sequences with one country, but some European sequences have no specific national information, for instance 496RUS26, so we have to use continents. In addition, some misspellings have been corrected, for instance “Filand”.
5) In Fig. 1 many even minor differences are perceptible in such groups of sequences as "R. flavescens". I suppose that this comes from low-quality sequences of an insufficient check of chromatograms for these new sequences.
A: We recheck the new sequence to improve the quality of the sequence and eliminate the minor differences in Figure 1.
6) Fig. 1 again, long-branch anomalies are visible for the 2 "R.laevis KUO" sequences from Finland, and for "R. delica H21527 Tunisia", which are only explicable by a problem in the alignment or when concatenating the sequences for the multigene analysis. It should be checked and resolved from the alignment.
A: We re-aligned the sequences and manually adjusted some wrong sequences to solved the problem of long branches of "R.laevis KUO" and "R. delica H21527 Tunisia".
7) line 171 : "flavescens" does not mean "turning yellowish bodies when bruised", but just "becoming yellow" in Latin.
A: We have changed "turning yellowish bodies when bruised" to "becoming yellow" in Latin.
8) Line 306 : "MHHNU 31049, epitype" for R.japonicoidea : probably coming from an earlier version of the manuscript, no epitype to be designated for this new species.
A: We deleted MHHNU 31049 as a epitype for R.japonicoidea.
9) Fig. 1 again, "subsect. Pallidosporinae" instead of "subsect. Pallidosporinea"
A: We have replaced "subsect. Pallidosporinea" with "subsect. Pallidosporinae" in Figure 1.

Reviewer 4 Report (Previous Reviewer 4)
As I performed the previous review, I consider improvement of the reviewed paper as very shallow and vague. To my disappointment, the authors used a lot of shortcuts, which raised questions about their approach and ethics.
1. In the re-drawn picture of R. flavescens (Fig. 5) authors showed different basidia contents, however the shape of basidia is used as before. Same applies for R. longicollis (fig. 8) where cystidia content in D, E is different while shape is exactly the same. How do authors explain that they used the same cystidia, but with different content? Drawings of the R. japonicoidea are not the same quality as the rest of the species. Moreover, with current drawings all elements appear to be thick-walled instead of thin-walled as stated in their description.
2. Authors ignored calls to improve typos and small errors, they left them as it is despite specifically requests in my previous review (Chinses in line 6, palldosporinae in lines 223, 470)
3. I specifically asked to use more recent literature while discussing morphological concepts and give as example R. chloroides. Authors did not change a word or values in the sentence "Russula chloroides (Krombh.) Bres. differs from R. longicollis in a longer stipe (60−80 mm), greenish at the top of stipe, larger hymenial cystidia up to 130 × 12 μm, and filiform, flexuous-undulate terminal cells in the pileipellis [9,11]". Instead of properly improving that sentence, they only put one more recent reference.
4. Authors left unattended, that R. brevipedum was originally raised under name R. brevipes in 2015 as indicated in MycoBank. Again, I specifically asked to correct that in my previous review.
5. How could authors explain the significantly different topology of R. pseudojaponica and R. flavescens?
Additional remarks
Authors should put higher effort in naming of their species. Naming them R. pseudojaponica and R. japonicoidea and putting same word etymology does not sound as exhausting effort.
Discussion is still rudimentary, focused on R. japonica. Why are you not discussing the general picture of R. subgen. Brevipedum in China?
Why did you remove the identification key?
References needs deep revision
Authors should consider the effort given in revision of their paper and improve paper respectively.
Author Response
Dear reviewer,
First, thank you very much for your valuable and specific comments on my paper. I am so sorry for ignored errors in previous paper. In this revision, we made more efforts to revise the paper. The specific amendments and explains are as follows:
- In the re-drawn picture of R. flavescens (Fig. 5) authors showed different basidia contents, however the shape of basidia is used as before. Same applies for R. longicollis (fig. 8) where cystidia content in D, E is different while shape is exactly the same. How do authors explain that they used the same cystidia, but with different content?
A: In Figure 5 and 8, we used PS software to draw and improve cystidia contents on the basis of the previous drawings in order to get a more realistic shape of the contents.
- Drawings of the R. japonicoidea are not the same quality as the rest of the species. Moreover, with current drawings all elements appear to be thick-walled instead of thin-walled as stated in their description.
A: In the revision, we re-draw the drawing of R. japonicoidea with coarser lines and rest species with thinner lines, so that they are the same quality.
- Authors ignored calls to improve typos and small errors, they left them as it is despite specifically requests in my previous review (Chinses in line 6, palldosporinae in lines 223, 470)
A: We corrected all typos and errors in the paper.
- I specifically asked to use more recent literature while discussing morphological concepts and give as example R. chloroides. Authors did not change a word or values in the sentence "Russula chloroides (Krombh.) Bres. differs from R. longicollis in a longer stipe (60−80 mm), greenish at the top of stipe, larger hymenial cystidia up to 130 × 12 μm, and filiform, flexuous-undulate terminal cells in the pileipellis [9,11]". Instead of properly improving that sentence, they only put one more recent reference.
A: We have supplemented more recent relevant literature when discussing morphological differences with similar species.
- Authors left unattended, that R. brevipedum was originally raised under name R. brevipes in 2015 as indicated in MycoBank. Again, I specifically asked to correct that in my previous review.
A: We re-introduced R. subg. brevipedum in line 40 and 41 of revison. “Russula subg. Brevipedum, typified by R. brevipes Peck, was originally described in 2015 as R. subg. Brevipes, but this was an invalid name and was changed to Brevipedum in 2020”
- How could authors explain the significantly different topology of R. pseudojaponica and R. flavescens?
A: As we added a mtSSU sequence (OP856854) of R. pseudojaponica, we get a different topology of R. pseudojaponica and R. flavescens.
- Authors should put higher effort in naming of their species. Naming them R. pseudojaponica and R. japonicoidea and putting same word etymology does not sound as exhausting effort.
A: “R. japonicoidea” was replaced by “R. brevispora”
- Discussion is still rudimentary, focused on R. japonica. Why are you not discussing the general picture of R. subgen. Brevipedum in China?
A: We have improved the discussion and compared the descriptions of some species of the subgenus in China with those in Europe and North America.
- Why did you remove the identification key?
A: The new key to has been added.
- References needs deep revision
A: We have improved, supplemented and updated the references.
In addition, we have corrected all the errors highlighted in your previous comments.

Round 2
Reviewer 3 Report (New Reviewer)
I appreciate the deep efforts of the authors to improve their manuscript and to make the discussion and bibliography more accurate. In this form and unless small details in the phylogeny would require a deeper attention (regarding sequence and alignment quality, about which the authors do not answer), no flagrant mistake could be found by me in this revised version, but one.
The authors still did not understand that "Russula littoralis Pegler" (a synonym of R. cremeolilacina) was not a Brevipedes. The one they really want to compare to their species is Russula littoralis Romagn. (MycoBank # 322934). Update accordingly lines 547, and key line 636 if based on Pegler's description.
Once these corrections made, and ref. 49 corrected too (as "Llistosella, J.; Pérez-de-Gregorio, M.À .; Llorens-Van-Waveren, I.", I am favourable to its publication.
Author Response
Dear Reviewer,
Thank you very much for your further suggestions. We will make changes and explanations to these suggestions as follows:
- In this form and unless small details in the phylogeny would require a deeper attention (regarding sequence and alignment quality, about which the authors do not answer).
First, the quality of the chromatograms of these new sequences are excellent (There were no miscellaneous peak). We removed the sequence beginnings and endings to ensure their accuracy. The sequences matrices were aligned using the MAFFT, and misalignments manually were adjusted in BioEdit. We can ensure that the quality of our sequences and alignments is unproblematic and phylogenic tree is trustworthy.
- The authors still did not understand that "Russula littoralis Pegler" (a synonym of R. cremeolilacina) was not a Brevipedes. The one they really want to compare to their species is Russula littoralis Romagn. (MycoBank # 322934). Update accordingly lines 547, and key line 636 if based on Pegler's description.
A: The paper compares "Russula littolaris" species based on Romagn's description, and "Russula littoralis Pegler" has been changed to “Russula littoralis Romagn.”
- 49 corrected too (as "Llistosella, J.; Pérez-de-Gregorio, M.À .; Llorens-Van-Waveren, I."
A: The reference 49 has been corrected.

Reviewer 4 Report (Previous Reviewer 4)
Paper was significantly improved. At this point I have only minor corrections
Lines 142 - 148 - This paragraph should be in Methods
Figure 2 - it is not clear to me why fig 2H has different scale bar. I propose to unified all scale bars to 20 (just delete half of the bar)
Line 208 - margin not cracked - you mean margin is even?
Line 232 - what mean colorless? you mean without crystaline content? check this in every description
Line 252 - would you please check SV reaction for every species once again? It is quite unusuall that such intricate reaction is different in closely related species
Line 255 - Pinus ? use full name of the genus
Line 268 - improve Error reference was not found
line 311 - would you please check once again spores dimensions? From microscopic picture it appears that spores are at least subglobose, but given Q express wide range starting from globose. Were all measured spores properly oriented in microscope? It is very common mistake, that not properly oriented spores are measured and then Q is much more lower.
Line 568 - Paragraph should start with full generic name Russula
Line 570 - list those 6 described species
Author Response
Dear reviewer,
Thank you very much for being able to provide us with nuanced and professional comments again. We have further improved our paper based on these comments.
Lines 142 - 148 - This paragraph should be in Methods
A: This paragraph has been moved to Methods.
Figure 2 - it is not clear to me why fig 2H has different scale bar. I propose to unified all scale bars to 20 (just delete half of the bar)
A: Figure 2, the scale of 2H has been adjusted to 20 mm.
Line 208 - margin not cracked - you mean margin is even?
A: “margin not cracked” means pileus margin is complete.
Line 232 - what mean colorless? you mean without crystaline content? check this in every description
A: “colorless” means color presented by the hyphae or cells in KOH solution. Should it be changed to “hyaline”?
Line 252 - would you please check SV reaction for every species once again? It is quite unusually that such intricate reaction is different in closely related species.
A: We re-checked SV reaction for every species, and corrected the results in the revision.
Line 255 - Pinus? use full name of the genus
A: Yes, I've complemented it with a full name.
Line 268 - improve Error reference was not found
A: Reinsertion of the references enabled them to be found.
line 311 - would you please check once again spores dimensions? From microscopic picture it appears that spores are at least subglobose, but given Q express wide range starting from globose. Were all measured spores properly oriented in microscope? It is very common mistake, that not properly oriented spores are measured and then Q is much more lower.
A: We observed and measured spores from holotype of Russula longicollis again. All observed spores were properly oriented (hilar appendix could be visualized). Finally, we obtain Q=(1.01–)1.04–1.07–1.11(–1.15). Spores mainly are subglobose (72.5%, Q>1.05), sometimes globose (27.5%, Q≤1.05).
Line 568 - Paragraph should start with full generic name Russula
A: This has been corrected.
Line 570 - list those 6 described species
A: The six described species have been listed.

This manuscript is a resubmission of an earlier submission. The following is a list of the peer review reports and author responses from that submission.
Round 1
Reviewer 1 Report
The manuscript entitled “Three new species of Russula subg. Brevipedum (Russulales, Russulaceae) from China”presented detailed morphological and phylogenetic studies of three proposed new species of Russula. However, I have to reject it in consideration of the scope and depth of your manuscript is not able to meet the requirement of this journal for here only presented three new species and lacked a discussion section and that not means that your results get wrong. I look forward to your resubmission after an improvement on following questions:
Language
Your manuscript needs careful editing and particular attention to English grammar, the italic format of species name, and sentence structure.
Introduction:
(1) In this part, it is suggested that the author should elaborate more fully on the information related to the genus, Russula, rather than eliciting subgenus information at the beginning of this article which might cause confusing.
(2) In paragragh 2, the ecological, biogeographic, medical information and former studies conducted in China were presented. This may seem like just a list of above informations, but it doesn’t reveal the research gap or the reason you started this study. This parts needs to be deeply dug and solved to improve the depth of this article. It seems to me that your article has changed my understanding of the species once called “Russula japonica”, which might be a cut-in point to explore.
Material and methods
Additional information about combining multi-locus dataset would be ideal.
Taxonomy
The ecology and distribution of proposed new species should be listed here.
Illustrations
(1) It is better to provide clearer SEM photograph of R. flavescens and R. longicollis;
(2) Scale bars could be shown on a figure more than once for there were some inaccuracy in current version.
Reviewer 2 Report
The authors describe three new species, but the taxonomic status of R. pseudojaponica and R. flavescens is not clearly elucidated. R. pseudojaponica and R. flavescens are phylogenetically and morphologically close to each other, the authors should clarify their differences for the two aspects in tail. However, in the current manuscript, none nucleotide differences and only a very tiny morphological difference have been described. It's a bit of a concern if they're two species. In addition, although the research content of this article conforms to the journal category, which is a little bit of simple and does not meet the level of JoF very well.
Reviewer 3 Report
Very good work. Only minor comments are made in the text. Many species without italics and other minor details.
The proposed species present a set of characters that support the hypothesis as new species.

Reviewer 4 Report
General appearance: Paper itself is properly structured, but not formatted according to Journal of Fungi Guidelines. I have found several cases of reference misapplication, where authors support their statements despite the fact that such information or idea was not mentioned in the original work. On top of that, the manuscript contains numerous mistakes, typos and errors, raising questions about overall precision of the conducted research. Manuscript needs deep and thorough revision to remove those mistakes.
Introduction is too general. I am completely missing the overview of the subgen. Brevipedum diversity at least in Asia. The statements are too broad and could be reduced to simpler sentences. I am missing some hypothesis or aims of the study as well. The last sentence belongs to the results, not aims.
Methodology is quite rudimentary, not properly explaining the dataset and obtaining results. Some relevant sequences and species are omitted in the analyses. This part has to be significantly improved and analyses should be recalculated with improved dataset according to the suggestions. Also, I specifically request to be provided with the alignment to see and revise differences between two of the described species (see my further comments).
Results: Some of the statements in the Results are not supported by phylogenetic analyses. In the species descriptions, authors discussed out-of-date literature and they are avoiding more recent studies.
Discussion: I am missing relevant discussion. Authors discussed only morphological differences in short notes under 3 described species. I expect some discussion about their ecology vs rest of the subgen. Brevipedum species, differences in tree topology in other works etc.
References: some of the references mentioned in the manuscript did not appeared in the List of used references (e.g. Caboň et al. 2017, Buyck 1989) or different reference is cited (e.g. Kornerup & Wanscher 1978 in text vs Kornerup & Wanscher 1981 in list of references)
More in-depth points (numbers refers to the lines):
Formatting is missing throughout the whole manuscript. Improve references according to the standards used in JoF.
Latin names should be italicized. This mistake runs throughout the whole manuscript.
6 - Chinese
27-28 - only one work can be valid in an infrageneric classification establishment. Why are 3 studies cited? Authors should use Mycobank.org to verify which publication is relevant to infrageneric names
34-35 - same question, moreover subgenus was described in different publications in 2015.
44-48 - is there something else left? better to simplify that members of this subgenus occured globally in all type of the forested habitats
48-49 - somewhere in the manuscript, you should mentioned species which were reported from China
66 - which magnifications did you use for observations? Did you use oil-immersion?
76- how was the cited protocol improved?
81 - Did you use this amplification protocol for all markers?
86 - Were PCR products somehow purified or not?
89 - Did you manually improve sequencing errors before submission to the genbank?
91- 92 - there are more sequences available in the genbank, which were available in those times. Why did you exclude them?
91 - it is not clear what was your sampling strategy in presenting ITS tree and multi-locus. Why did you use 3 additional sequences of R. cremicolor? Some of the species are missing additional markers but ITS. Why did you not put effort and loan or collect additional Chinese species to build proper multi-locus? Also, I am missing recently described asian species e.g. R. callainomarginis, but there are more ommited species.
Fig. 1 - What is the meaning to show a separate ITS tree, since you have used ITS in multi-locus as well?
139-140 - I would hesitate to recognize R. flavescens and R. pseudojaponica as two separate species since R. flavescens is in polytomy. I would like to see alignments first.
148 - R. flavescens -diagnosis length of the basidiospores is highly variable and could be affected by multiple environmental factors (e.g. thicker hummus or fallen leaves layer) therefore is not appropriate to use that as main discriminating character. Especially, when your field observations are based on 4 specimens.
197, 199, 357 Palldosporinae
R. pseudojaponica - in the etymology, you stated that "“pseudojaponica” refers to morphologically similar to R. japonica", however in diagnosis "Russula pseudojaponica is similar to R. flavescens"
289 - you should use more recent literature to discuss morphological differences. Bon 1988 is definitelly not appropriate for R. chloroides.
In general, paper needs deep revision in all its aspects. Authors should carefully consider whether each reference corresponds to the statement provided by authors. Main discriminating morphological differences among species should be visualized in form of some table. Molecular analyses should be repeated with inclusion of the relevant sequences.
I will check and comment on morphological descriptions in another round of the revision.